# Hydrogen peroxide serves as pivotal fountainhead for aerosol aqueous sulfate formation from a global perspective

Jie Gao [1,2], Haoqi Wang [1,2], Wenqi Liu[1], Han Xu[1], Yuting Wei[1], Xiao Tian[1], Yinchang Feng [1], Shaojie Song [1] ✉ & Guoliang Shi [1] ✉

Traditional atmospheric chemistry posits that sulfur dioxide ($SO_2$) can be oxidized to sulfate ($SO_4^{2-}$) through aqueous-phase reactions in clouds and gas-phase oxidation. Despite adequate knowledge of traditional mechanisms, several studies have highlighted the potential for $SO_2$ oxidation within aerosol water. Given the widespread presence of tropospheric aerosols, $SO_4^{2-}$ production through aqueous-phase oxidation in aerosol water could have a pervasive global impact. Here, we quantify the potential contributions of aerosol aqueous pathways to global sulfate formation based on the GEOS-Chem simulations and subsequent theoretical calculations. Hydrogen peroxide ($H_2O_2$) oxidation significantly influences continental regions both horizontally and vertically. Over the past two decades, shifts in the formation pathways within typical cities reveal an intriguing trend: despite reductions in $SO_2$ emissions, the increased atmospheric oxidation capacities, like rising $H_2O_2$ levels, prevent a steady decline in $SO_4^{2-}$ concentrations. Abating oxidants would facilitate the benefit of $SO_2$ reduction and the positive feedback in sulfate mitigation.

Sulfur dioxide ($SO_2$) has been on the radar by numerous regions, due to the repercussions for air quality and acidic deposition[1,2]. Through control measure deployment, global $SO_2$ emissions have remarkably decreased since the 1980s[3]. However, the corresponding product, atmospheric sulfate ($SO_4^{2-}$), has declined more slowly than $SO_2$ emissions in the continental regions, across North America, Europe, and Asia, implying the complexity of its formation process[4]. Till now, $SO_4^{2-}$ is still a major component of fine particle matter ($PM_{2.5}$), impacting the global radiation budget, climate change, and public health[5,6].

The efforts of deep dive into the sulfate formation mechanisms are continually ongoing. The classic chemical mechanisms including gas-phase oxidation of $SO_2$ by hydroxyl radical (OH) and a suite of in-cloud aqueous oxidation pathways (the main oxidants contain hydrogen peroxide ($H_2O_2$), nitrogen dioxide ($NO_2$), ozone ($O_3$), and oxygen ($O_2$) catalyzed by transition metal ions, TMI) have been well recognized and included in most large-scale atmospheric chemistry models[7]. The global three-dimensional chemical transport model simulations indicated that gas-phase oxidation contributed about 20% to tropospheric sulfate[8], while aqueous-phase reactions accounted for about 80% of the global sulfate production rate[9]. The studies on in-cloud aqueous pathways demonstrated that oxidation of $SO_2$ was mainly due to the reaction with $H_2O_2$ on a global scale[8]; the role of $O_3$ oxidation in sulfate production has enhanced (up to 17–27%) in recent years[6]; TMI-catalyzed S(IV) oxidation could account for 9–17% of total sulfate production globally[10].

Nonetheless, for sulfate concentration, the gap between traditional model simulations and observations still existed on a regional scale, especially during severe pollution in winter[11], thus deriving

[1]State Environmental Protection Key Laboratory of Urban Ambient Air Particulate Matter Pollution Prevention and Control, Tianjin Key Laboratory of Urban Transport Emission Research, China Meteorological Administration-Nankai University Cooperative Laboratory for Atmospheric Environment-Health Research, College of Environmental Science and Engineering, Nankai University, Tianjin 300350, China. [2]These authors contributed equally: Jie Gao, Haoqi Wang ✉e-mail: songs@nankai.edu.cn; nksgl@nankai.edu.cn

scientific interest in potential sulfate formation mechanisms other than cloud chemistry, like aerosol-mediated multiphase and heterogeneous chemistry[12]. Although the content of aerosol liquid water is smaller than that of cloud water[13], aerosols are ubiquitous in the free troposphere, which can also provide a potential environment and contribute to sulfate formation[14].

Various oxidation reaction mechanisms in aerosols have been proposed in the exploration of unconsidered pathways of sulfate formation and rapid accumulation under contaminated conditions, like haze events in some industrial-intensive areas[12,15]. Aerosol aqueous-phase oxidation (including $H_2O_2$, $NO_2$, $O_3$, and TMI pathways) is considered to play an essential role, due to the evidence that the missing sulfate content (the difference between measured and simulated sulfate) is closely related to aerosol water content (AWC)[16]. The significance of aqueous-phase oxidation has also been validated by direct evidence through ambient isotopic observations[17,18]. Besides, in some specific situations, $SO_2$ in the atmosphere can also undergo heterogeneous oxidation on surfaces of black carbon[19], mineral dust[20], soot particles[21], and aqueous-phase aerosols[22]; photochemical oxidations like in-particle nitrate photolysis[23] and atmospheric photosensitization[24] are also considered to play active roles in sulfate formation. However, large uncertainties of the kinetic parameters still existed, such as rate constants and uptake coefficients[12,15]. Although these mechanisms are likely to be notable in some regions, whether the specific reaction conditions of these mechanisms are satisfied when extrapolating to the worldwide atmospheric environment is still pending.

Considering that the key role of aerosol water has been widely acknowledged, existing studies have further evaluated the relative importance of four aerosol aqueous-phase pathways ($H_2O_2$, $NO_2$, $O_3$, and TMI) in specific local regions. For instance, reactive nitrogen chemistry in aerosol water has been considered to be a source of sulfate during haze events in northern China, under favorable conditions with high $NO_2$, aerosol water, and pH (which is used to characterize aerosol acidity)[16]; and regime transitions of the aerosol aqueous-phase pathways in the North China Plain highly depended on both pH and oxidants/catalysts[25]. However, scientific knowledge on the relative importance of aerosol aqueous pathways at a global level is extremely limited[12]. Seeing that the temporal and spatial distributions of $SO_2$, oxidants, catalysts, and aerosol characters vary greatly worldwide, the expected dominant pathways would be variable in different regions and periods. Thus, a global comprehensive simulation would be imperative to assess aerosol aqueous-pathway contributions to offset the lack of comparability globally.

In this work, we elucidate the significance of aqueous-phase sulfate formation pathways (i.e., $H_2O_2$, $NO_2$, $O_3$, and TMI) in aerosol water based on a series of theoretical calculations utilizing fundamental data provided by the GEOS-Chem chemical transport model. We focus on the global spatiotemporal variabilities for January, April, July, and October 2019, both surface and vertical spatial scales. We also investigate the temporal trends of aqueous sulfate formation pathways and the corresponding influencing factors in typical cities over 2001–2019 based on the simulations in 2001, 2005, 2009, 2013, 2017, and 2019. The aqueous-phase pathways in aerosol water could make undeniable contributions to sulfate formation globally. The abundance of the regional dominant oxidant, like $H_2O_2$, can be the key control. The understanding of sulfate formation regimes on global and regional scales can be useful for performing atmospheric oxidant control, implementing corresponding reduction policies, and further achieving sulfate mitigation.

## Results
### Regime of aerosol aqueous sulfate formation pathways at surface level
Disparate chemical regimes and reaction pathways of sulfate formation in aerosol water prevailed in different regions and seasons,

depending on meteorological conditions, aerosol properties, and reactant concentrations (Fig. 1, Supplementary Figs. S1–S4). The simulation indicated that most of the continental surface areas (approximately 80%) were dominated by $H_2O_2$ oxidation. $H_2O_2$ is very soluble in aqueous phase and is one of the most effective oxidants of S(IV). The near pH independence of the $H_2O_2$ oxidation reaction favors more chances for sulfate formation over a wide range of pH, which is caused by the fact that the pH dependences of the S(IV) solubility and the reaction rate constant cancel each other[7]. The importance of $H_2O_2$ pathway in aerosol multiphase chemistry is also supported by several recent studies[12]. An experimental research has demonstrated sulfate formation rate through $H_2O_2$ oxidation can be enhanced by the high ionic strength of aerosol particles (Supplementary Fig. S5)[26], and field observation suggests that aerosol phase $H_2O_2$ concentration can be higher than that predicted from partitioning of gaseous phase $H_2O_2$[27]. Furthermore, the isotope analysis also highlights the vital role of $SO_2$ oxidation by $H_2O_2$[28].

Aided by the increase in the solubility of transition metals under low aerosol pH conditions, the availability of catalysts in the aqueous phase would increase[12,15]. $O_2$ oxidation catalyzed by soluble TMI, like Mn(II) and Fe(III), would play a central role in sulfate formation over the regions with higher aerosol acidity and sufficient catalysts, particularly in parts of North Africa, West Asia, South Asia, and Oceania, notably in April and July (Supplementary Fig. S6). Several isotopes and modeling studies also indicate a major role of TMI-catalyzed oxidation during haze episodes[29,30]. However, a recent study focused on East Asia has suggested the declined acidity in the future would weaken the role of TMI-related sulfate formation in the downstream areas of dust sources[31]. The dominant role of TMI pathway in the corresponding regions might be more susceptible to reduced anthropogenic acid precursor emissions and increased alkaline substance emissions, due to the acid-driven solubilization of TMI[13].

The atmospheric importance of the $NO_2$ pathway in aerosol particles has been debated in the last few years. $NO_2$ oxidation could be essential when accompanied by high concentrations of $NO_2$ and neutralizing species at high aerosol pH. In this simulation, the ascendancy of $NO_2$ oxidation was generally not evident on a global scale, except for a small part of North America and East Asia in January. Consistent with the past study[16], $NO_2$ pathway would become pivotal only when the aerosol pH was above 5, which was generally greater than the majority of worldwide pH values inferred from thermodynamic modeling[32]. Even under similar pH conditions, the $H_2O_2$ pathway also appears to have opportunities for oxidation, because of the pH-independence of reaction rate[7]. $NO_2$ pathway was less competitive compared with other pathways globally. Moreover, an isotopic study also suggested that the contribution of $NO_2$ pathways would not cause the enrichment of [34]S in sulfate[28], which was in line with another work based on $\Delta^{17}O_{sulfate}$ observations and GEOS-Chem model that declared the minor role of $NO_2$ pathway[30].

The $O_3$ oxidation showed strong effects in more alkaline regions, as $O_3$ can react most rapidly with sulfite ion ($SO_3^{2-}$), which equilibrium concentrations would be improved under higher pH conditions[7]. The extent of the regions dominated by the $O_3$ pathway was mainly consistent with the distribution of deserts. The Sahara Desert area represented the most influence by $O_3$ oxidation due to the higher alkalinity, especially in January and April. The contribution of the $O_3$ pathway also showed seasonality in other deserts, including the Taklimakan Desert, the Arabian Desert, and the Australian Desert. The importance of the $O_3$ pathway may be increased, as the expected aerosol acidity continues to decline because of the projected reduction of acidic components and possible growth of anthropogenic ammonia ($NH_3$) emission[32].

Overall, $H_2O_2$ oxidation was found to play the dominant role in aerosol aqueous sulfate formation on a global scale. TMI and $O_3$ pathways were of paramount importance in some localized areas (high

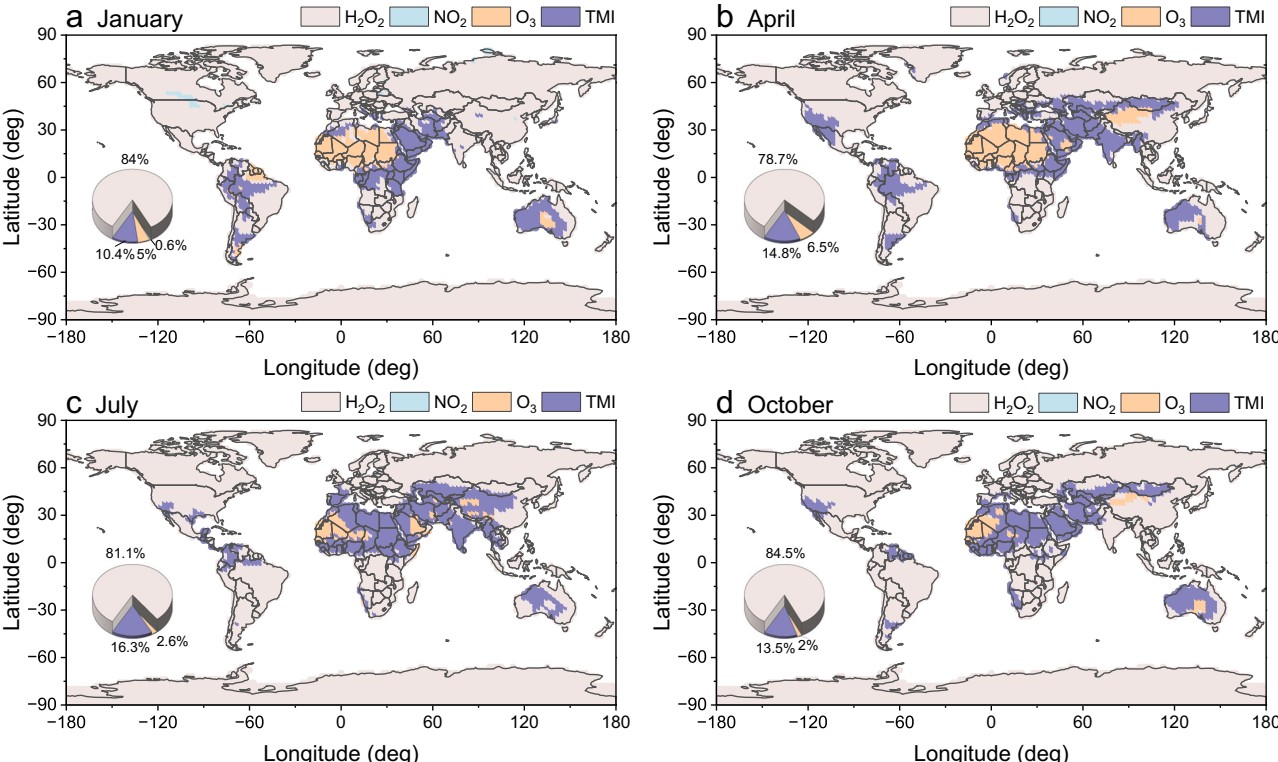

**Fig. 1 | Predominant sulfate formation pathways in aerosol water at the surface level in 2019. a** January, **b** April, **c** July, and **d** October. The aqueous-phase pathway ($H_2O_2$, $NO_2$, $O_3$, and TMI) with the highest oxidation rate are colored in the corresponding area. The pie charts show the proportions dominated by different oxidation pathways on the surface in continental areas. Globally, $H_2O_2$ oxidation was the major pathway, though TMI and $O_3$ pathways could also carry a great deal of weight in sulfate formation in some regions and seasons. Source data are provided as a Source Data file.

acidity and high alkalinity, respectively). The global distribution of elevated $SO_4^{2-}$ concentration almost matched the high contribution of summed aqueous oxidations, both remarkable in South Asia, East Asia, and West Asia (Supplementary Fig. S7). Besides, our calculations showed that the contribution of aerosol aqueous-phase oxidation was comparable to that of the OH gas-phase oxidation (Supplementary Fig. S8).

**Vertical profile of aerosol aqueous sulfate formation pathways**
Figure 2 shows the global vertical distribution of the sulfate production pathways in aerosol water organized by latitude in January and July 2019 (details in Supplementary Figs. S9 and S10). Consistent with the surface results, the pathway distribution gave evidence that all four pathways contributed to sulfate formation considerably and were highly dependent on aerosol acidity, oxidants, and catalysts, at 950 mbar in January. The cleaner regions at high latitudes were dominated by $H_2O_2$ oxidation. The primary pathway at 40°–60°N remained $H_2O_2$ oxidation, but $NO_2$ oxidation could also contribute roughly 10–20% due to the relatively high $NO_2$ concentration in these areas (Supplementary Fig. S11g). The $O_3$, TMI, and $H_2O_2$ pathways exhibited competition in the low and middle-latitude regions of the Northern Hemisphere in January (Fig. 2b). Elevated Mn(II) and Fe(III) concentrations at these latitudes (Supplementary Fig. S11k, S11m) contributed to the dominance of the TMI pathway. The essential role of $O_3$ oxidation pathway could be attributed to the higher $O_3$ levels and higher aerosol pH observed in specific regions. In the remaining regions from the equator to the mid-latitudes of the Southern Hemisphere, the TMI pathway dominated due to the favorable Mn(II) and Fe(III) concentrations and pH conditions in these areas.

As the atmosphere tends to become cleaner at higher altitudes, the reduction in $SO_2$ and oxidant concentrations would lead to a corresponding decline in $SO_4^{2-}$ production rate. In January, the formation rates at 900 mbar and 800 mbar near the top of the boundary layer were slightly reduced from those at 950 mbar (Fig. 2b). The relative contribution of formation pathways was almost similar to that at 950 mbar, despite the relative contribution of $H_2O_2$ had a slight increase at most latitudes. With decreasing atmospheric pressure in the vertical direction right along, the $SO_4^{2-}$ formation and its dominant pathways started to change more obviously, caused by the variations in the physicochemical properties of aerosols and other influencing factors in the vertical dimension. The sulfate formation rates at 600 mbar were 1-2 orders of magnitude lower than that at 950 mbar. At this altitude, in addition to $H_2O_2$ oxidation, the TMI pathway also played a role in middle-latitude regions, since the aerosol acidity was sufficiently low, which ensured the availability of catalysts (Supplementary Fig. S11a). At 400 mbar, although aerosol acidity was greater, $O_3$ oxidation still dominated over other pathways in the tropical regions of the Northern Hemisphere owing to the increase of $O_3$ abundance (Supplementary Fig. S11i); the regions at other latitudes were prevailed by $H_2O_2$ oxidation. Since a zonally-averaged profile of sulfate production rates indicated that few sulfates existed at lower pressure[8], no further discussion was undertaken below 400 mbar.

In July, $SO_2$ concentrations decreased to lower levels compared with that in January, but rather than declining, sulfate generally increased especially in the middle latitudes of the Northern Hemisphere (Supplementary Fig. S12). Moreover, global aerosol acidity in July overall increased compared with that in January, especially in the Northern Hemisphere (Supplementary Fig. S13). Due to the elevated acidic conditions that can increase the dissolution of transition metal ions, the concentrations of Mn(II) and Fe(III) in aerosol water were raised in the mid-latitude regions of the Northern Hemisphere (Supplementary Fig. S11l, S11m). This phenomenon resulted in extremely

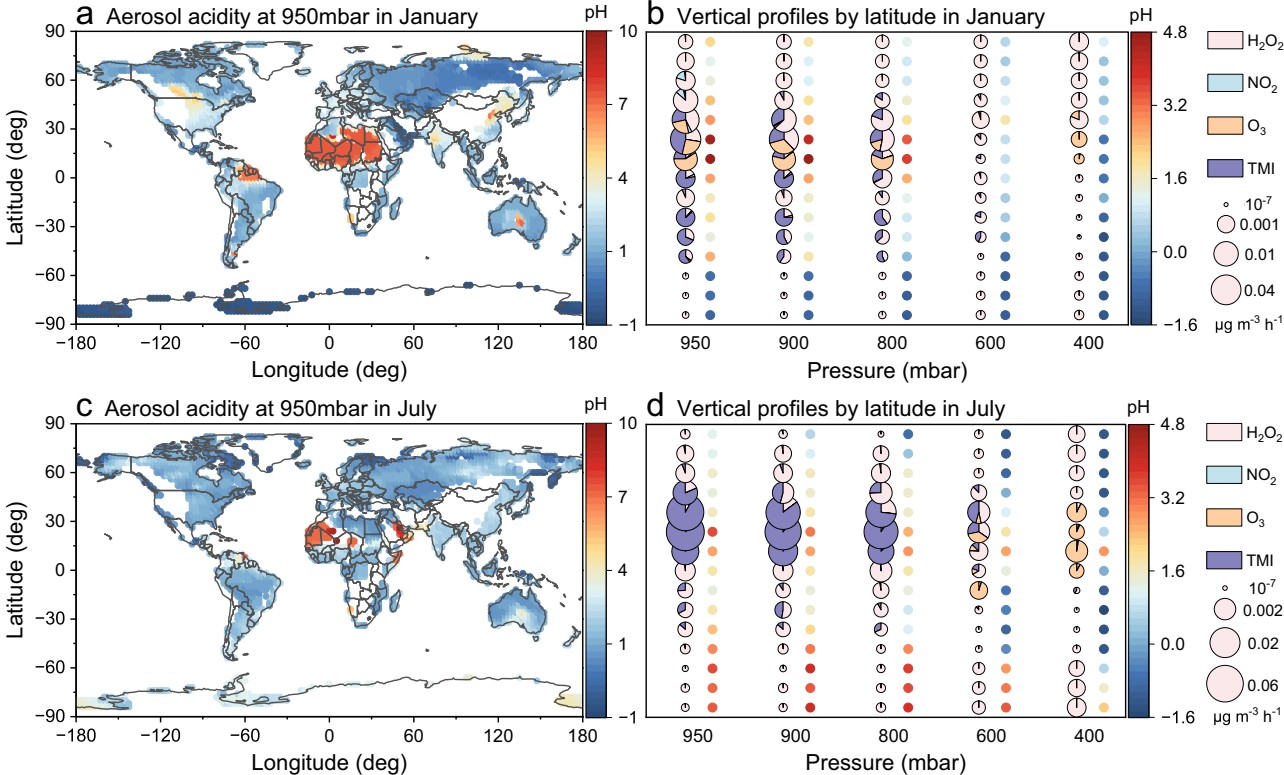

**Fig. 2 | Global distribution of aerosol acidity at 950 mbar and vertical profiles of aqueous sulfate formation pathways average by latitude in 2019. a, b** January and **c, d** July. The pie sizes display the total rate of sulfate productions (average by latitude), and the pie colors represent the contributions from the corresponding formation pathways ($H_2O_2$, $NO_2$, $O_3$, and TMI). The colored dots to the right of the pies indicate the mean pH values at that latitude. The vertical profiles of formation pathways showed seasonal features and they were distinct at different atmospheric pressures. Source data are provided as a Source Data file.

dominant oxidation through TMI pathway with high sulfate production rates over other pathways in mid-latitude regions, obviously in multiple isobars, like 950, 900, and 800 mbar (Fig. 2d). As the altitude continues to increase in the mid-latitude regions, $H_2O_2$ and $O_3$ pathways started to play key roles at 600 mbar; $O_3$ oxidation would be the predominant pathway at 400 mbar. The regions at other latitudes were almost regulated by $H_2O_2$ oxidation.

In brief, sulfate formation rates within the boundary layer were consistently substantial and subsequently decreased with the rise of altitude. $H_2O_2$ oxidation still showed a vital role in aerosol aqueous sulfate formation as altitude changes, while the potential contributions of TMI and $O_3$ pathways could not be overlooked in middle latitude regions.

### Temporal trend of aerosol aqueous sulfate formation in typical cities

Over the past two decades, the majority of the selected city areas (with disparate geographic locations, meteorological conditions, and atmospheric pollution levels) have exhibited a declining trend of $SO_4^{2-}$ concentrations in January, with $H_2O_2$ oxidation emerging as the primary production pathway (Fig. 3). Notably, urban areas in North America and Europe such as Washington DC, Los Angeles, and London consistently maintained comparatively low sulfate levels, often below 5 μg m$^{-3}$. Meanwhile, urban regions in East Asia, including Beijing, Shanghai, and Hong Kong, exhibited the most substantial reduction in sulfate concentrations. Although there was an overall decreased trend in aerosol acidity globally, this alone was not enough to offset the fact that the TMI pathway emerged as the second most important route after $H_2O_2$ oxidation. This was evident vertically in cities such as Shanghai, Hong Kong, Los Angeles, and London (Supplementary Fig. S14). Besides, the Beijing region exhibited an apparent trend of

increasing aerosol pH, which is likely to create favorable conditions for the $NO_2$ oxidation and result in the non-negligible contribution of the $NO_2$ pathway in this specific area. Meanwhile, Canberra, situated in the Southern Hemisphere, maintained remarkably stable aerosol acidity, sulfate concentrations, and dominant pathways over the past 20 years.

Understanding the city-specific sulfate formation pathways in aerosol water is crucial for deciphering broader implications and guiding future strategies for targeted sulfate mitigation. The interannual fluctuation of $SO_2$, oxidants, catalysts, and AWC would together impact the change of the four oxidation pathways and also sulfate concentration in typical urban areas over the past two decades. The actual determinants of sulfate decline diverged. To address these distinctions, we explored the change rate of pathway contributions and the change rate of its influencing factors (subsequently normalized), specifically for each year compared with 2001. To better demonstrate the relationship between changing trends in pathways and factors, we introduced the relative incremental contrast (RIC), i.e., the change rate of each factor divided by the change rate of pathway contribution. This approach enabled us to evaluate their drivers using consistent criteria (Supplementary Figs. S15–S18).

Sulfate concentrations in Chinese cities showed two-step declines from 2001 to 2009 and 2013 to 2019 (Fig. 3), which was closely associated with the fluctuation of $H_2O_2$ pathway contribution and its relevant factors. The previous step could be attributed to the decrease in aerosol water content (Fig. 4a). By providing more medium for multiphase reactions, increased AWC has been proven to accelerate secondary aerosol formation and trigger the positive feedback between water uptake and secondary formation in haze development, enhancing light scattering and reducing visibility, as well as suppressing the boundary layer height and worsening air pollutions[33,34]. Compared with 2001, $SO_2$ emissions were enhanced in both 2005 and 2009, and

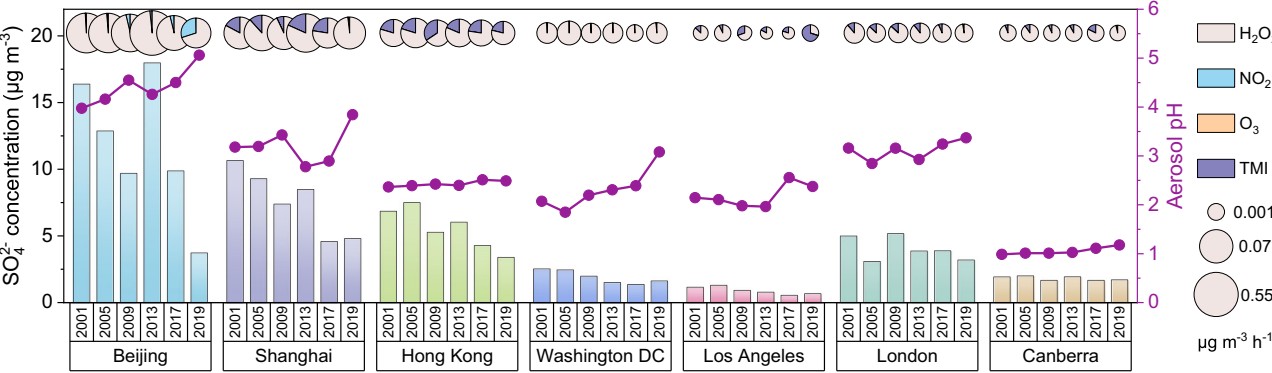

**Fig. 3 | Temporal trends in sulfate concentrations, aerosol pH, and aerosol aqueous pathways in selected urban areas in January from 2001 to 2019.** The bar charts represent sulfate concentrations (left axis). The dot-line charts represent aerosol pH (right axis). The pie charts represent the sulfate formation rate of four aqueous-phase pathways in the aerosol water, distinguished by color. Source data are provided as a Source Data file.

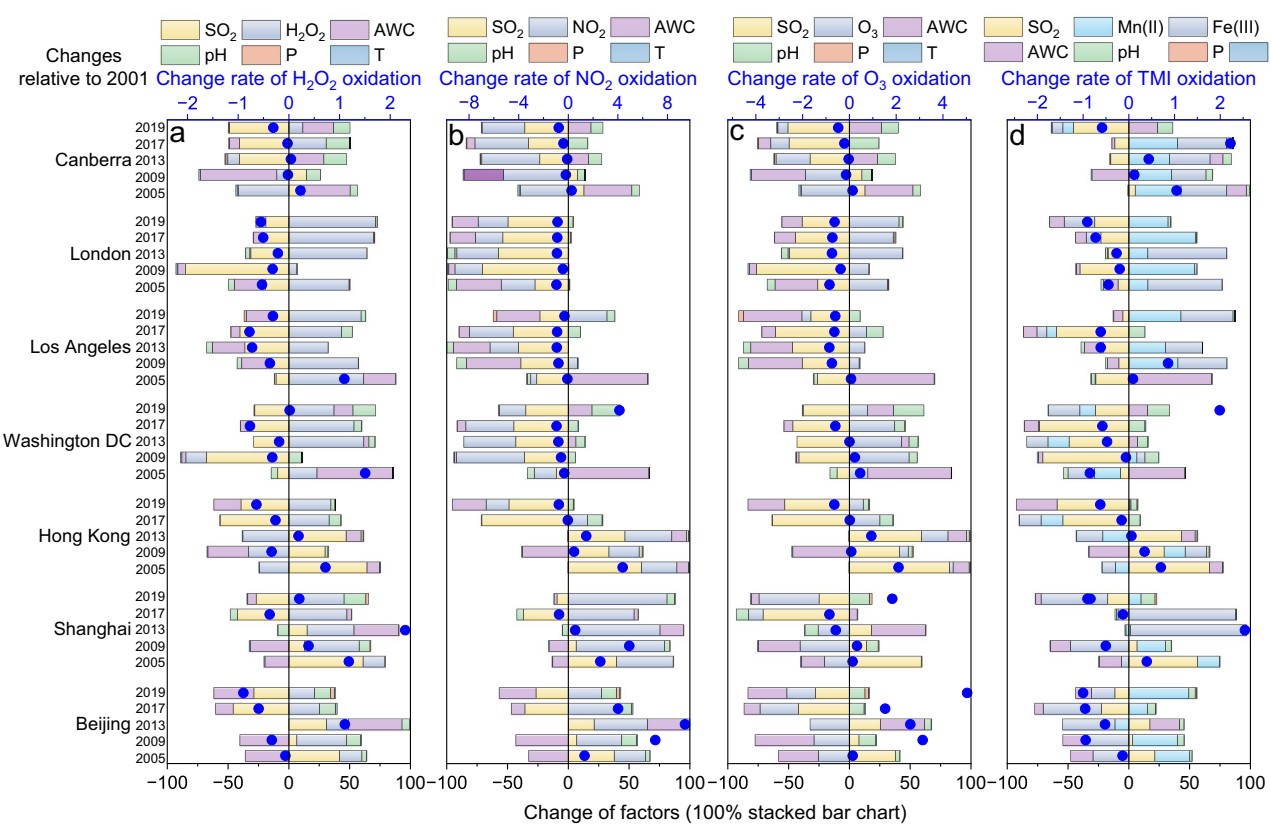

**Fig. 4 | Temporal trends in oxidation rates and influencing factors for aerosol aqueous sulfate formation pathways in selected urban areas in 2005, 2009, 2013, 2017, and 2019, compared with that in 2001. a** $H_2O_2$ pathway, **b** $NO_2$ pathway, **c** $O_3$ pathway, and **d** TMI pathway. The dot charts represent the change rate of oxidation rates (upper axis). The bar charts indicate the change of the influencing factors, shown by 100% stacked bar charts (lower axis). Source data are provided as a Source Data file.

the concentrations of the vital oxidant $H_2O_2$ were also increased (Fig. 4a). Nevertheless, $SO_2$ lacked an effective oxidation medium due to the reduced AWC[16,35]. The positive benefits of the reduced AWC outweighed the negative impacts of $SO_2$ emissions and $H_2O_2$ concentrations, ultimately leading to lower sulfate concentrations during this period. The situation worsened in 2013 with conditions of high $SO_2$ emissions, high $H_2O_2$ concentrations, and high AWC, leading to a rebound in sulfate concentrations (Fig. 3). Following a series of measures implemented by China to control precursor emissions, there was a substantial decrease in $SO_2$ emissions[36,37]. This sulfate reduction observed during 2017 and 2019 could be attributed to these measures.

However, $H_2O_2$ did not return to the 2001 level, which implied that oxidants could be a crucial factor that restricted the ongoing decrease of regional sulfate concentrations in these urban regions. The $H_2O_2$ pathway in selected cities in Europe and North America almost decreased in oxidation rate over the past years relative to 2001. Notably, this might be more due to reducing the precursor $SO_2$ as the increased $H_2O_2$ played the opposite role (Fig. 4a). We further designed sensitivity experiments with $H_2O_2$, $SO_2$, pH, and AWC fixed for 2001 to investigate the drivers of changes in the $H_2O_2$ pathway. The results confirmed that elevated $H_2O_2$ concentrations over the past two decades inhibited rapid sulfate reduction in these urban areas, and in

some years, nearly offset the positive benefits of $SO_2$ reduction (Supplementary Fig. S19). Additionally, the role of AWC could not be ignored owing to the positive correlation with $H_2O_2$ oxidation in most cases; the effect of pH changes could be less important because the $H_2O_2$ pathway was insensitive to pH variation, though acidity variation takes active roles in other aqueous pathways[7].

Alongside $H_2O_2$ oxidation, it's crucial to highlight the significance of the TMI pathway, which played a pivotal role in some selected cities. Though TMI pathway in these years almost showed a downward trend compared with that in 2001, the attention should also be directed towards the elevated concentrations of Mn(II) and Fe(III), especially in Los Angeles, London, and Canberra (Fig. 4d). Moreover, despite the less importance of $NO_2$ and $O_3$ pathways, we also found some commonalities in pathway changes. The contributions from $NO_2$ and $O_3$ pathways generally declined in Washington DC, Los Angeles, London, and Canberra relative to 2001, and the change rates remained relatively stable. Conversely, there were significant interannual differences and apparent increases in Beijing, Shanghai, and Hong Kong, which is likely to be caused by the combined effects of change in the precursor, oxidants, pH, and AWC, wherein the increasing $NO_2$ levels and raising aerosol pH would take responsibility for the growth of $NO_2$ pathway; the raising aerosol pH would also be one of the non-negligible reasons for the increase in $O_3$ oxidation (Fig. 4b, c).

## Discussion

Worldwide regimes of sulfate formation in aqueous aerosols vary in horizontal and vertical directions. The regional emission characteristics and seasonality take effect in pollutant concentration, aerosol acidity, and formation pathway profiles. In brief, though the TMI pathway and $O_3$ pathway also occupied a crucial position in some districts under appropriate reaction conditions, $H_2O_2$ oxidation was the predominant pathway in most continental areas. Indeed, there could also be prospects that other aerosol-mediated reaction mechanisms reported recently could play key roles in sulfate formation, especially in polluted environments. Besides, uncertainties and deviations in the simulation of aerosol aqueous oxidations still exist due to the complexity of atmospheric multiphase chemistry. For instance, the important effects of aerosol ionic strength on certain reactions have been reported by experimental studies, like enhancing $H_2O_2$ oxidation rate while inhibiting TMI-catalyzed oxidation rate in deliquesced aerosol particles[26], enhancing the multiphase oxidation rate of $SO_2$ by $O_3$ in aqueous acidified sea salt aerosols[38], elevating the reaction rate constants for the oxidation of S(IV) by $NO_2$ in the aerosol water[39]. Despite prior studies, more knowledge of the kinetics and thermodynamics in high ionic strength solutions is still needed for establishing solute strength-dependent kinetics, like tighter modeling and experimental constraints on kinetic parameters at different aerosol acidities and ionic strengths under conditions closer to the real atmosphere[11,12].

The complex pathways of sulfate formation and the variety of influencing factors pose a challenge to the development of effective control policies at low $SO_2$ levels. Given that the continuous decline of $SO_2$ emission is largely achieved on a global scale, attention to other driving factors is highlighted, like atmospheric oxidation capacity, especially in different regions. Although it's undeniable that $SO_2$ reduction is the most fundamental and significant influence for sulfate abatement[3,40], the negative feedback of the increased $H_2O_2$ concentration has been verified in North America[41] and East Asia[42], and cannot be ignored in the context of the increased oxidizing capacity of the earth's atmosphere. Affected by the changes in tropospheric chemistry in recent years owing to the increase of carbon monoxide (CO), nitric oxide (NO), nitrogen dioxide ($NO_2$), volatile organic compounds (VOCs), as well as the increasing UV-B (ultraviolet radiation b) radiation caused by the depletion of stratospheric ozone, the evidence from ice core samples drilled at Summit Greenland showed a 60%

increase in $H_2O_2$ concentrations during the last 150 years[43]; a prediction evaluated with a one-dimensional photochemical mode also suggested that $H_2O_2$ increase from 1980 to 2030 could be 100% or more in the urban boundary layer[44].

As a vital photochemical secondary product, $H_2O_2$ generated from the binding of two $HO_2$ radicals can serve as an oxidant both in their own right and as a reservoir species for $HO_x$ (OH and $HO_2$) radicals. OH can be produced via the photolysis of $O_3$, nitrous acid ($HNO_2$), and aldehydes. $HO_2$ can be formatted through the photo-oxidation of CO and VOCs by the OH radical, degradation of formaldehyde (HCHO) and other aldehydes by photolysis or by reaction with OH radical, the decomposition of peroxyacetyl nitrate (PAN), and the photodegradation of aromatic hydrocarbons[45]. It's worth noting that the photochemical formation of $H_2O_2$ through $HO_2$ could be sensitive to ambient NO levels, due to the reaction of NO with $HO_2$ being faster than the bimolecular combination of $HO_2$, which could lead to substantial suppression of $H_2O_2$ formation via $HO_2$ if NO is abundant over a hundred ppt[45]. This limitation effect has been reported in some areas, like Jungfraujoch Observatory in Switzerland[46], Brittany in France[47], and Hong Kong in China[48]. However, elevated $H_2O_2$ mixing ratios were also observed in some regions even during winter haze events with high NO and low $HO_2$ levels, like North China Plain[49], implying other sources. Besides, observed $H_2O_2$ in the particle phase based on some urban sampling (Los Angeles, Beijing) was much greater than the concentration predicted by gas-particle partitioning from Henry's law, also indicating that the capability of generating $H_2O_2$ in aerosols, like redox chemistry of complexed transition metals and other redox-active species[27,50]. For instance, field observations and laboratory experiments have proved that the photochemistry of $H_2O_2$ in-particle formation can be driven by transition metal ions (TMIs) and humic-like substances (HULIS) in deliquescent aerosols (RH > 50%)[49]. Photochemical aging of atmospheric fine particles was also proved as a potential source for gas phase $H_2O_2$ under relatively dry conditions during daytime[51]. Another noteworthy source of gaseous $H_2O_2$ is the ozonolysis of alkene that is independent of radiation, which may be prominent in low photochemical conditions and may serve as the potential tropospheric source during autumn and winter at mid or high latitudes over the continents[52,53] like Pabstthum in Germany (nighttime)[54], Guangzhou in China (nighttime)[55], and Mt. Tai in China (nighttime)[56]. Overall, $H_2O_2$ concentration in the atmosphere is dependent on the pollutant levels (CO, NO, $NO_2$, VOCs, $O_3$, TMI) and meteorological parameters (light intensity, temperature, water vapor content), intimately linking to $O_3$ and $HO_x$ cycles, reflecting oxidation capacity of the troposphere. Given the close interlink between $H_2O_2$, $NO_2$, and $O_3$, the fluctuation of a certain oxidant is likely to affect other correlated oxidant levels, leading to indirect pathway contributions to sulfate production. Unraveling the interactive relationship between the precursors is also key to better understanding the sulfate formation mechanism. Further studies on the response of $O_3$ and $H_2O_2$ to $NO_x$ (NO + $NO_2$) and VOCs as well as the interaction between secondary photochemical oxidants and aerosols are greatly needed, especially at various spatiotemporal scales. Attention to relevant anthropogenic emissions including fossil fuel combustion, industrial processes, vehicle exhaust, mineral dust, and biomass burning is imperative and control measures should be on the agenda, such as improving energy efficiency and supplying clean energy.

Despite pollutant emissions (like $SO_2$, $NO_x$, and VOCs) declining, some secondary particles and key factors (such as oxidation capacity, acidity, and toxicity) have not decreased to the same extent, implying insufficient budget estimation relative to the sources and sinks[57]. Sulfate formation could be the right example that there might be unidentified pathways, unconsidered mechanisms, or unappreciated sources of oxidants, making the lower reduction than $SO_2$. Oxidant sources are gradually becoming a key factor in the limitation of further sulfate decline and may have a greater inhibitory effect in the future.

The benefits of reducing atmospheric oxidation capacity and local key-pathway-relevant oxidant levels, like $H_2O_2$, can facilitate the benefit of $SO_2$ reduction and thus optimize sulfate abatement.

## Methods

### GEOS-Chem model simulation and ISORROPIA II model calculation

The global three-dimensional GEOS-Chem chemical transport model (version 13.3.4)[58] is used to simulate the gaseous pollutant and aerosol components in 2019. To investigate temporal trends, we also conducted simulations in 2001, 2005, 2009, 2013, 2017, and 2019, and selected Beijing, Shanghai, Hong Kong, Washington DC, Los Angeles, London, and Canberra as representative cities for detailed discussion, considering the varied geographical locations, meteorological conditions, and air pollution levels. The meteorological inputs are obtained on the Modern-Era Retrospective Analysis for Research and Applications, version 2 (MERRA-2) product from the Goddard Earth Observing System (GEOS) of the NASA Global Modeling and Assimilation Office (GMAO)[59]. A coarser horizontal resolution of 5° (longitude) ×4° (latitude) is applied in the GEOS-Chem simulation as the initial and boundary conditions for the nested-grid 2.5° (longitude) ×2° (latitude) simulation. The vertical grid contains 47 pressure levels from the surface to the mesosphere.

The mass concentrations of some species are estimated based on the accumulation mode sea salt (SALA) and the accumulation mode dust (DST$_1$). Na$^+$ = SALA × 39.63%[60,61], Cl$^-$ = SALA × 60.85%[60,61], K$^+$ = SALA × 1.1%[60,61], Ca$^{2+}$ = DST$_1$ × 3%[62,63], Mg$^{2+}$ = DST$_1$ × 0.36%[63], Mn = DST$_1$ × 0.063%[25,64], Fe = DST$_1$ × 2.425%[25,64]. The solubilities of trace metals are intended to be from 5% to 50% for Mn and from 0.45% to 1% for Fe[30]. Although there may be some uncertainties, here, 5% of Mn is assumed to be dissolved, with 100% in the form of Mn(II); and the solubility of Fe is assumed to be 0.45%, with 50% in Fe(III) oxidation states[10,30]. The concentrations of Mn(II) and Fe(III) (in mol L$^{-1}$) in aerosol water can be calculated as follows[25]:

$$Mn(II) = Min(C_{Mn} \cdot FS_{Mn(II)}/AWC, 1.6 \times 10^{-13}/[OH^-]^2) \quad (1)$$

$$Fe(III) = Min(C_{Fe} \cdot FS_{Fe(III)}/AWC, 2.6 \times 10^{-38}/[OH^-]^3) \quad (2)$$

where $C_{Mn}$ and $C_{Fe}$ represent the concentration of Mn and Fe in fine particle components, mols per liter of air; $FS_{Mn(II)}$ and $FS_{Fe(III)}$ are the fractional solubilities of Mn(II) and Fe(III), respectively; AWC is the aerosol liquid water content, liters per liter of air.

The aerosol pH, aerosol water content (AWC), and ionic strength (I) play vital roles in sulfate aqueous formation, which are calculated by the ISORROPIA II model, a widely used thermodynamic equilibrium model with a high computational efficiency (http://isorropia.epfl.ch)[65]. The model is run in the "forward mode" and "metastable state". Aerosol pH (Supplementary Fig. S6) and ionic strength (Supplementary Fig. S5) are calculated.

$$pH = -\log_{10} \frac{1000 H^+_{air}}{AWC} \quad (3)$$

where $H^+_{air}$ is the H$^+$ loading, μg m$^{-3}$; AWC is the aerosol water content, μg m$^{-3}$.

$$I = \frac{1}{2} \sum_{i=1}^{n} m_i z_i^2 \quad (4)$$

where I is the ionic strength, mol L$^{-1}$; n is the number of different ions in solution; $m_i$ is the molarity, mol kg$^{-1}$; $z_i$ is the number of ion charges.

For data calibration, the simulation results of PM$_{2.5}$, SO$_4^{2-}$/ PM$_{2.5}$, pH, AWC, and gaseous pollutants in this work are verified by comparing with the reanalysis datasets or observation-constrained datasets. The global reanalysis dataset "Satellite-derived PM$_{2.5}$" from the Atmospheric Composition Analysis Group at Washington University in St. Louis (https://sites.wustl.edu/acag/datasets/surface-pm2-5/, last access: 2024-3-31) is utilized (Supplementary Figs. S20, S21). We also compare the concentration ratio of SO$_4^{2-}$/PM$_{2.5}$ between GEOS-Chem model simulations and field observations or reanalysis datasets (Supplementary Figs. S22, S23), by gathering the publicly available data from the CHAP reanalysis dataset for China (https://weijing-rs.github.io/product.html), the EBAS observation network in European countries (https://ebas-data.nilu.no/Default.aspx), and the IMPROVE observation network in the United States (https://views.cira.colostate.edu/fed/Express/ImproveData.aspx). Aerosol pH (Supplementary Figs. S24, S25), AWC, and gaseous pollutants are verified by observation-constrained data from previous studies. The relevant datasets for reference and validation can be referred to https://doi.org/10.6084/m9.figshare.24967032.

### Sulfate formation through aqueous-phase oxidation in aerosol water

Multiphase oxidation of $SO_2$ in aerosol water mainly involved three parts, namely, the transport of $SO_2$ and oxidants (i.e., $H_2O_2$, $NO_2$, $O_3$, and $O_2$) into the condensed aerosol phase, the dissolution of hydrated $SO_2$ and oxidants in the aqueous phase following Henry's law, and aqueous oxidation of S(IV) (hydrated $SO_2$ ($SO_2 \cdot H_2O$) + bisulfite ion ($HSO_3^-$) + sulfite ion ($SO_3^{2-}$)) into $SO_4^{2-}$[12].

The mass transfer rate coefficient $k_{MT}$ (s$^{-1}$) of $SO_2$ or oxidants can be determined by[7,16]:

$$k_{MT}(X) = \left[ \frac{R_p^2}{3D_g} + \frac{4R_p}{3\alpha v} \right]^{-1} \quad (5)$$

where $R_p$ is the mean radius of aerosol particles, m; $D_g$ is the gas-phase molecular diffusion coefficient, m$^2$ s$^{-1}$; α is the mass accommodation coefficient on droplet surface; and v is the mean molecular speed, m s$^{-1}$.

The gas-aqueous equilibrium for $SO_2$ or oxidants can be calculated by the Henry's law[7]:

$$[X(aq)] = H_{T_0} \exp\left[ -\frac{\Delta H_{298K}}{R} \left( \frac{1}{T} - \frac{1}{T_0} \right) \right] \times P_{X(g)} \quad (6)$$

where [X(aq)] is the aqueous concentration of X species, mol L$^{-1}$; $H_{T_0}$ is Henry's constant at 298K, mol L$^{-1}$ atm$^{-1}$; T is a specific temperature, K; $T_0$ is 298 K; and $P_{X(g)}$ is the partial pressure of X in the gas phase, atm.

The oxidation rate of different pathways in aerosol water can be expressed as[7,18,66]:

$$R_{H_2O_2 + S(IV)} = k_1[H^+][HSO_3^-][H_2O_2(aq)]/(1 + K[H^+]) \quad (7)$$

$$R_{NO_2 + S(IV)} = (k_2/[H^+] + K_{S2}k_3/[H^+]^2)K_{S1}[SO_2 \cdot H_2O(aq)][NO_2(aq)] \quad (8)$$

$$R_{O_3 + S(IV)} = k_4[SO_2 \cdot H_2O] + k_5[HSO_3^-] + k_6[SO_3^{2-}])[O_3(aq)] \quad (9)$$

$$R_{TMI + S(IV)} = \begin{cases} k_7[H^+]^{-0.74}[S(IV)][Mn(II)][Fe(III)], pH \leq 4.2 \\ k_8[H^+]^{0.67}[S(IV)][Mn(II)][Fe(III)], pH > 4.2 \end{cases} \quad (10)$$

where $R_{oxidant+S(IV)}$ represents the chemical reactions rate in aqueous phase, mol $L^{-1}$ $s^{-1}$; $k_1$ - $k_8$ are the reaction rate constants, $M^{-1}$ $s^{-1}$ or $M^{-2}$ $s^{-1}$; $K_{s1}$ and $K_{s2}$ are the thermodynamic dissociation constants of $SO_2 \cdot H_2O$, mol $L^{-1}$; and [X] is the aqueous concentration, mol $L^{-1}$.

The final rate of sulfate production by a certain oxidant in aerosol aqueous water $P_{oxidant+S(IV)}$ ($\mu g$ $m^{-3}$ $h^{-1}$) can be determined by:

$$P_{oxidant+S(IV)} = 3600\,s\,h^{-1} \cdot 96\,g\,mol^{-1} \cdot \frac{L_a}{\rho_w} \cdot R_{H,\,oxidant+S(IV)} \tag{11}$$

where 3600 s $h^{-1}$ is a time conversion factor; 96 g $mol^{-1}$ is the molar mass of $SO_4^{2-}$; $L_a$ is aerosol water content, mg $m^{-3}$; $\rho_w$ is the water density, 1 kg $L^{-1}$; and $R_{H,\,oxidant+S(IV)}$ is the overall reaction rate, mol $L^{-1}$ $s^{-1}$.

More details are referred to the Supplementary Information Text S1–S4.

**Temporal trend calculation of oxidation pathway and influence factor**
Compared with 2001, the change rates of oxidation rate for the four pathways and the change rates of their corresponding influencing factors in 2005, 2009, 2013, 2017, and 2019 were computed to assess the temporal trend of four aqueous pathways and analyze the role of the influence factors in oxidation pathways. Taking 2001 as the target enabled us to evaluate the variation using consistent criteria.

$$\text{Change rate of oxidant}_j\text{ pathway in year}_i = \frac{P_{oxidant_j\,in\,year_i} - P_{oxidant_j\,in\,year_{2001}}}{P_{oxidant_j\,in\,year_{2001}}} \tag{12}$$

where $P_{oxidant_j\,in\,year_i}$ represents the sulfate production rate ($\mu g$ $m^{-3}$ $h^{-1}$) by oxidant$_j$ pathway (including $H_2O_2$, $NO_2$, $O_3$, and TMI) in a specific year$_i$, i.e., 2005, 2009, 2013, 2017, and 2019; $P_{oxidant_j\,in\,year_{2001}}$ represents the production rate ($\mu g$ $m^{-3}$ $h^{-1}$) by the oxidant$_j$ pathway in 2001.

$$\text{Change rate of influencing factor}_k\text{ in year}_i$$
$$= \frac{factor_k\,in\,year_i - factor_k\,in\,year_{2001}}{factor_k\,in\,year_{2001}} \tag{13}$$

where factor$_k$ in year$_i$ represents the values of the influencing factor$_k$ in a specific year$_i$, i.e., 2005, 2009, 2013, 2017, and 2019, including the concentration of $SO_2$ (ppb), $H_2O_2$ (ppb), $NO_2$ (ppb), $O_3$ (ppb), Mn(II) (mol $L^{-1}$), Fe(III) (mol $L^{-1}$), AWC ($\mu g$ $m^{-3}$), and the values of T (K), P (Pa), pH. The factor$_k$ in year$_{2001}$ represents the values of the influencing factor$_k$ in 2001. The changes in the influencing factors for each pathway in Fig. 4 and Supplementary Figs. S15–S18 are shown by 100% stacked bar charts.

The previous study has applied the relative incremental response (RIR) as the change of the $SO_4^{2-}$ production rate in response to the change in oxidant or $SO_2$ emissions[67]. Similarly, we define the relative incremental contrast (RIC) here to examine the potential relationships between the temporal trend (compared with 2001) of oxidation rates and influencing factors.

$$\text{RIC between factor}_k\text{ and oxidant}_j\text{ pathway in year}_i$$
$$= \frac{\text{Change rate of influencing factor}_k\text{ in year}_i}{\text{Change rate of oxidant}_j\text{ pathway in year}_i} \tag{14}$$

A positive RIC value represents the consistent change trend of oxidation pathways and influencing factors. Larger values of RIC may suggest a more important role of the influencing factors. The RIC

between four aqueous pathways and their driving factors are displayed in Supplementary Figs. S15–S18, shown by 100% stacked bar charts.

**Reporting summary**
Further information on research design is available in the Nature Portfolio Reporting Summary linked to this article.

## Data availability
The authors declare that the main data supporting the findings of this study are available within the article, its Supplementary Information file, and the Source data. Source data are provided with this paper, and can be accessed online via the Figshare https://doi.org/10.6084/m9.figshare.24967032. The global reanalysis dataset "Satellite-derived $PM_{2.5}$" was obtained from the Atmospheric Composition Analysis Group at Washington University in St. Louis (https://sites.wustl.edu/acag/datasets/surface-pm2-5/). The reanalysis datasets of sulfate and $PM_{2.5}$ for the China region were obtained from the ChinaHigh-AirPollutants (CHAP) dataset (https://weijing-rs.github.io/product.html). The field observation datasets of sulfate and $PM_{2.5}$ were collected from the EBAS observation network in European countries (https://ebas-data.nilu.no/Default.aspx) and the IMPROVE observation network in the United States (https://views.cira.colostate.edu/fed/Express/ImproveData.aspx). Data analysis and draw designs were conducted based on Matlab R2021b, Excel 2016, and Origin 2024. The maps within images were prepared using the built-in shapefiles in Origin software. We do not contain third-party images. Source data are provided with this paper.

## Code availability
ISORROPIA-II is available at: http://isorropia.epfl.ch. GEOS-Chem is available at: https://doi.org/10.5281/zenodo.5764874.

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

## Acknowledgements

This study was supported by the National Natural Science Foundation of China (42205109 to S.S. and 42077191 to G.S.), the National Key Research and Development Program of China (2023YFC3709502 to G.S. and 2022YFC3703400 to G.S.). S.S. also acknowledged support from the Natural Science Foundation of Tianjin City (22JCYBJC01330) and TianHe Qingsuo open research fund of TSYS in 2022 & NSCC-TJ.

## Author contributions

J.G. and H.W. contributed equally to this work. S.S., G.S., and Y.F. designed the research. J.G. and H.W. performed research. W.L., H.X., Y.W., and X.T. helped with model simulations. J.G. and H.W. analyzed and interpreted model results. J.G., H.W. S.S., and G.S. drafted the manuscript.

## Competing interests

The authors declare no competing interests.
