## [Peer Review File · Nature Communications]

Hydrogen peroxide serves as pivotal fountainhead for aerosol aqueous sulfate formation from a global perspectiveReviewer #1 (Remarks to the Author):

Review of "Hydrogen peroxide: Pivotal fountainhead for aerosol aqueous sulfate formation as 2 inferred from global perspective" by Gao et al. The manuscript used GEOS-Chem global atmospheric chemistry transport model to understand contributions of sulphate formation in aqueous phase via different pathways. The simulation and analyses are focused on both global picture and some of megacities to illustrate that H₂O₂ is the major pathway of SO₂ aqueous oxidation which contribute largely into the particulate sulphate formation and hence air pollution. The authors suggests that, despite of continuous decrease of SO₂ emission, the increased atmospheric oxidation capacity (as indicated by the rising H₂O₂) prevents a steady decline in sulphate. Based on their results, authors advice that further measures to intervent and reduce the atmospheric oxidation capacity would be the key to further improve air quality. While the importance of multiphase formation and aqueous formation of sulphate and oxidation of SO₂ have been widely discussed in previous modelling, observational and experimental studies, providing a global picture of relative importance of different pathways and insights into the historical trend of changing of drivers of SO₄ formation is new. However, authors would need to more carefully demonstrate the robustness of their simulation results and discuss about the relevant uncertainties, in order to deliver a convincing conclusion and suggestion to policymaking. While I will detail my major concerns and a few minor suggestions below, I would like to express my general support for this interesting piece of work, and I recommend its publication in Nature Communication once my comments are addressed.

1) It is a nice modelling study, as authors correctly and clearly introduced in the introduction that it is still poorly understand about the aqueous phase SO₂ formation, leading to large uncertainty in air quality and PM modelling. I think if authors could demonstrate that after adding in these aqueous oxidation pathways, GEOS-Chem model does improve the performance in SO₄ simulation, this will greatly improve the robustness of this work. Furthermore, I would like to suggest more rigorous references to and comparison against previous observational studies would be a great help to convince the audience that conclusion from this study is sounding and also help to understand the uncertainty of modelling analysis. For example, as mentioned above already, how is the performance of SO₄ simulation get improved? How is the simulated aerosol water and pH compared with observations in the chosen cities? There are plenty of published observational datasets are available and greatly helpful with the discussion.

2) Authors mention in the abstract, and many other places, that rising H₂O₂ has prevented a steady decline in SO₄. This may not be true from the observations. For example, in Beijing especially during heavy haze episode, large reduction in SO₂ emission has led to the change of dominated inorganic from sulphate to nitrate (Wang et al., 2020) and references therein. During these polluted episodes, although not dominate by sulphate, very high aerosol liquid water has been observed in Beijing up to 80 ug/m³. Not only Beijing, New Delhi also observed recorded high aerosol water up to 260 ug/m³ (Chen et al., 2022), which could contribute largely to aqueous phase SO₂ oxidation.

3) Line 90-102. I feel it is more belong to the results. And here suddenly discuss about MBL sulphate is a bit distracting. I also do not understand that how could you infer recent 20 years conditions from simulation 2019?

4) Both NO₂ and O₃ show strong effect in high pH, and typically NO₂ and O₃ are highly correlated with each other. How do authors disentangle the effect of them? I suspect that if you turn off oxidation pathway of one of them, the contribution from the other one will increase, due to completion with the SO₂ oxidation. This might show some indirect contributions. However, this is just curious, H₂O₂ is the dominant factor anyhow.

5) Line 168. Four modes? I do not understand it.

6) Line 240. Attributed to favourable meteorological conditions? Do authors expect that the meteorological conditions change largely between 2013-2019 and 2001-2009? If yes, please provide evidence.

7) There are 7 megacities are chosen for more detailed analysis. Why these cities are chosen? I guess it is because of the availability of observational datasets from these cities and reasonable validation of the modelling outcomes. This links to my 1st point, more rigorous reference to previous observational studies are needed.

8) line 289. "negative feedback of the increased H₂O₂". And line 310, authors suggest for controlling of H₂O₂. H₂O₂ is not primarily emitted, it comes from secondary formation which is highly non-linear. So, more detailed explanation of why do we have the negative feedback of the increased H₂O₂, what are the major sources and pathways of H₂O₂, which primary emission sources should be targeted in the control. The authors suggest VOCs are on an upward trend. This may not be true, at least not everywhere, e.g., in UK VOCs do keep decreasing every year (<https://www.gov.uk/government/statistics/emissions-of-air-pollutants/emissions-of-air-pollutants-in-the-uk-non-methane-volatile-organic-compounds-nmvocs>). Further discussion and clearer conclusive summary will improve the value of the paper and make more explicit and doable suggestions for the policymakers.

Method: mainly for clarification and improve the reproducibility.

1) accumulation mode of sea salt and dust are considered in the study, however, they are more abundant in the coarse model which is not considered. Please justify this.

2) Line 424-423. Please provide the references for the metals speciation fractions.

3) Equation-4. Ionic strength. Where do you use it. I may missed that, but please explain after the eq.4, after you calculate the ionic strength, how do you use it for your analysis.

4) Eq.6. Please provide the values and references for the Henry's constants, and R value.

5) Eq.7-10. How do $k_1 \sim k_8$ been calculated? It seems not clear for me.

References:

Chen, Y., Wang, Y., Nenes, A., Wild, O., Song, S., Hu, D., Liu, D., He, J., Hildebrandt Ruiz, L., Apte, J. S., Gunthe, S. S., and Liu, P.: Ammonium Chloride Associated Aerosol Liquid Water Enhances Haze in Delhi, India, *Environmental Science & Technology*, 56, 7163-7173, 10.1021/acs.est.2c00650, 2022.

Wang, Y., Chen, Y., Wu, Z., Shang, D., Bian, Y., Du, Z., Schmitt, S. H., Su, R., Gkatzelis, G. I., Schlag, P., Hohaus, T., Voliotis, A., Lu, K., Zeng, L., Zhao, C., Alfarra, M. R., McFiggans, G., Wiedensohler, A., Kiendler-Scharr, A., Zhang, Y., and Hu, M.: Mutual promotion between aerosol particle liquid water and particulate nitrate enhancement leads to severe nitrate-dominated particulate matter pollution and low visibility, *Atmos. Chem. Phys.*, 20, 2161-2175, 10.5194/acp-20-2161-2020, 2020.

Reviewer #2 (Remarks to the Author):

General Comments:

In this study, the authors show a quantification of the potential contribution of the aerosol aqueous pathways to global sulfate production using the GEOS-Chem model. Results display that sulfate in most of the continental surface areas (approximately 80%) were derived from H₂O₂ oxidation. It's very amazing. This work highlights the equal importance of managing both oxidants and precursors for effective sulfate control. I think this manuscript can be published after minor revision. There are several comments as following:

Main Comments:

1. The oxidation of SO₂ by OH is an important pathway for sulfate formation. Globally, this

pathway contributes 20-27% to sulfate production (Alexander et al., 2009; Sofen et al., 2011). However, the OH pathway is not involved in GEOS-Chem model simulation. This pathway is expected to be added to their simulation.

2. In the introduction you mention the GEOS-Chem chemical transport model, but does that model address the above-mentioned areas of deficiency that you mention. This is puzzling. So the advantages of the model are not stated in the manuscript.

3. In this study, their results are obtained only from the GEOS-Chem model simulation. The credibility of information in this study is a bit low due to the absence of observation data.

4. This study focused on sulfate formation pathways in continental regions. The transport of marine oxidants, precursors and aerosols to continents can have influences on the pathways of sulfate formation. Could the authors consider the transport impacts on sulfate formation?

5. Sulfate formation pathways in the haze events are quite distinct from the clean days. Such, the sulfate formation pathways in a city with serious air pollution are quite distinct from those in a city with low levels of aerosol pollution. However, results in this paper show that the H₂O₂ pathway is dominated the production of sulfate in Beijing, Shanghai, Hong Kong, Washington, and London, etc. The levels of aerosol pollution in these cities are quite different, but the sulfate formation pathways are quite similar. So, how do the authors explain this phenomenon?

6. Some areas should be explained more clearly. For example, why TMI has a stronger core effect in areas of higher aerosol acidity. Whereas most of the continental surfaces are about 80% hydrogen peroxide oxidized. the dominance of NO₂ oxidation is not obvious. the characteristics of the regions dominated by TMI, H₂O₂, and NO₂ could have been described (should have been a bit more clear like the O₃ description).

7. Giving a general conclusion based on the regional characteristics. The third point is that in my opinion some of the more important diagrams could be put into the body of the text, such as some of the important drivers (S14-S17), and given their corresponding explanations.

8. In the introduction you mentioned that the main research focused on anthropogenic influences. However, there are relatively few analyses that mention anthropogenic activities in the discussion, so might you consider adding this part to the discussion. For example, involving anthropogenic activities in the future perspective at the end.

Reference

Alexander, B., Park, R. J., Jacob, D. J., Gong, S. 2009. Transition metal-catalyzed oxidation of atmospheric sulfur: Global implications for the sulfur budget. *Journal of Geophysical Research: Atmospheres*, 114(D2).

Sofen, E. D., Alexander, B., Kunasek, S. A. 2011. The impact of anthropogenic emissions on atmospheric sulfate production pathways, oxidants, and ice core $\Delta^{17}O$ (SO₄²⁻). *Atmospheric Chemistry and Physics*, 11(7), 3565-3578.

Response to reviews

We extend our appreciation to the two reviewers for their positive comments and interest in our manuscript. The insightful and constructive suggestions were invaluable in the reworking and improvement of this paper. We identified four common concerns from Reviewers 1 and 2, and we have addressed them in the revised manuscript. A summary is provided below.

First, we are careful in addressing the concerns of both Reviewers 1 and 2. We have added the model calibration and data comparison to verify the plausibility and robustness of our simulations in the revised manuscript. The datasets for reference and validation have been uploaded to the public data platform Figshare (currently accessible private link: <https://figshare.com/s/f4d4c43c4a3c516d1b2c>, activated link after paper publication: <https://doi.org/10.6084/m9.figshare.24967032>). In this version of the resubmission, we quantitatively calibrated the simulation results of the GEOS-Chem simulations using the global satellite-derived PM_{2.5} concentrations (<https://sites.wustl.edu/acag/datasets/surface-pm2-5/>), as well as aerosol acidity (pH) and aerosol liquid water content (AWC) publicly reported in recent papers. The results indicated that our simulations were acceptable. We also gathered validation datasets and analyzed the ratios of sulfate to PM_{2.5} for some focal regions, including China (CHAP, <https://weijing-rs.github.io/product.html>), European countries (EBAS, <https://ebas-data.nilu.no/Default.aspx>), as well as the United States (IMPROVE, <https://views.cira.colostate.edu/fed/Express/ImproveData.aspx>). These field observations/reanalysis datasets often yield higher values than the modeled results, affirming the potential significance of the aerosol water pathway in sulfate formation.

Another important piece of advice is the necessity of further discussion of limitations, uncertainties, and the implications for policymaking in this study. While a significant number of studies have explored the formation of sulfate aerosols from various perspectives, our study here assesses aerosol aqueous-pathway contributions to sulfate formation on a global scale. It is essential to acknowledge the uncertainties owing to the complexity of sulfate formation and the influence of some factors such as ionic strength, organic complexes, and catalyst valence. We further stress the importance of gaining an understanding of the kinetics and thermodynamics of high ionic strength solutions, particularly under conditions closer to the real atmosphere. We also combed through the main sources of the important oxidant (hydrogen peroxide, H₂O₂) in detail and highlighted concerns about the associated anthropogenic emission sources.

Moreover, we appreciate the inquiry regarding the assessment of the relative importance of the gas-phase pathway and the aerosol aqueous-phase pathway in sulfate formation. Undoubtedly, gas-phase oxidation via OH radical is the traditional

and widely known pathway for sulfate formation. In the revised manuscript, we have evaluated the formation rates of OH gas-phase oxidation and compared them with those of aerosol aqueous-phase oxidation in January, April, July, and October 2019. Our findings reveal that the OH radical pathway is comparable to the aerosol aqueous-phase pathway. Numerous studies have underscored the role of the OH pathway. Our study here stands out for highlighting the global significance of the aerosol aqueous-phase pathway.

We would also like to report that a new figure (Fig. 4) has been added to the main text, based on the recommendation of Reviewer 2. Fig. 4 underscores the trends in oxidation rates of H₂O₂, NO₂, O₃, and TMI pathways and their influencing factors in 2005, 2009, 2013, 2017, and 2019, compared with that in 2001. In these selected urban areas, precursors, oxidants, and aerosol water content are all important factors affecting sulfate formation rate changes.

A point-by-point response to the reviewers' comments has been provided below. The original comments from reviewers are in normal font, our replies are **indented and in blue**, and revised texts as they appear in the manuscript text are **indented and in red**.

Fig. 4 | Temporal trends in oxidation rates and influencing factors for (a) H₂O₂, (b) NO₂, (c) O₃, and (d) TMI pathways in selected urban areas in 2005, 2009, 2013, 2017, and 2019, compared with that in 2001. The dot charts represent the change rate of oxidation rates (upper axis). The bar charts indicate the change of the influencing factors, shown by 100% stacked bar charts (lower axis).

Point-by-point response to reviewers' comments

Reviewer #1

Review of “Hydrogen peroxide: Pivotal fountainhead for aerosol aqueous sulfate formation as 2 inferred from global perspective” by Gao et al. The manuscript used GEOS-Chem global atmospheric chemistry transport model to understand contributions of sulphate formation in aqueous phase via different pathways. The simulation and analyses are focused on both global picture and some of megacities to illustrate that H_2O_2 is the major pathway of SO_2 aqueous oxidation which contribute largely into the particulate sulphate formation and hence air pollution. The authors suggests that, despite of continuous decrease of SO_2 emission, the increased atmospheric oxidation capacity (as indicated by the rising H_2O_2) prevents a steady decline in sulphate. Based on their results, authors advice that further measures to intervene and reduce the atmospheric oxidation capacity would be the key to further improve air quality. While the importance of multiphase formation and aqueous formation of sulphate and oxidation of SO_2 have been widely discussed in previous modelling, observational and experimental studies, providing a global picture of relative importance of different pathways and insights into the historical trend of changing of drivers of SO_4 formation is new. However, authors would need to more carefully demonstrate the robustness of their simulation results and discuss about the relevant uncertainties, in order to deliver a convincing conclusion and suggestion to policymaking. While I will detail my major concerns and a few minor suggestions below, I would like to express my general support for this interesting piece of work, and I recommend its publication in Nature Communication once my comments are addressed.

Response: Thank you for taking the time to review our manuscript. We sincerely appreciate your positive and constructive comments that we have carefully addressed in the revised manuscript. We are pleased to hear that you found our study to be an interesting contribution to the understanding of sulfate formation in the aerosol aqueous phase on a global scale. We acknowledge your advice regarding the need for a more thorough demonstration of the robustness of our simulation results and a discussion of the relevant uncertainties. We are committed to addressing your concerns and making necessary revisions to improve the clarity of our manuscript. We have carefully addressed these points in our revised manuscript and provided a point-by-point response below.

1) It is a nice modelling study, as authors correctly and clearly introduced in the introduction that it is still poorly understood about the aqueous phase SO₂ formation, leading to large uncertainty in air quality and PM modelling. I think if authors could demonstrate that after adding in these aqueous oxidation pathways, GEOS-Chem model does improve the performance in SO₄ simulation, this will greatly improve the robustness of this work. Furthermore, I would like to suggest more rigorous references to and comparison against previous observational studies would be a great help to convince the audience that conclusion from this study is sounding and also help to understand the uncertainty of modelling analysis. For example, as mentioned above already, how is the performance of SO₄ simulation get improved? How is the simulated aerosol water and pH compared with observations in the chosen cities? There are plenty of published observational datasets are available and greatly helpful with the discussion.

Response: We appreciate your thoughtful advice. We have declared relevant uncertainties of the model simulations in the discussion as well as added the model calibration and data comparison to address the robustness in the revised manuscript. The model results perform well compared with the observational datasets; and the aerosol aqueous-phase pathways can indeed contribute to sulfate formation, which could be a potentiality for improving the performance of sulfate simulation.

For PM_{2.5}, we conducted calibration by utilizing the reanalysis dataset “Satellite-derived PM_{2.5}” from the Atmospheric Composition Analysis Group at Washington University in St. Louis (<https://sites.wustl.edu/acag/datasets/surface-pm2-5/>, last access: 2024-3-31). This dataset estimated global PM_{2.5} concentrations using satellite, model, and ground-based monitoring data. Aerosol optical thickness from satellites such as MODIS, VIIRS, MISR, and SeaWiFS are combined with ground-based solar photometer (AERONET) observations to produce geophysical estimates. Statistical fusion methods further integrate additional PM_{2.5} measurements. The results are illustrated in Fig. S20. The spatial distribution of PM_{2.5} concentration in four months of 2019 obtained from GEOS-Chem simulations closely resembled that of the reanalysis dataset “Satellite-derived PM_{2.5}”. Moreover, we focused on North Africa (5°N–30°N, 10°W–30°E), Oceania (10°S–45°S, 110°E–155°E), East Asia (5°N–58°N, 100°E–150°E), Europe (37°N–70°N, 9°W–34°E), and North America (25°N–65°N, 60°W–140°W) for further comparison (Fig. S21). Both the mean and median of GEOS-Chem simulations for these regions were acceptable, though slightly overestimated compared with the dataset.

For sulfate, the GEOS-Chem model utilized in this study accounted for sulfate formation just in the gas phase and cloud water phase. To validate the

significance of sulfate formation in aerosol liquid water, we compared the concentration ratio of $\text{SO}_4^{2-}/\text{PM}_{2.5}$ between model simulations and field observations/reanalysis datasets. We gathered the publicly available data from the CHAP reanalysis dataset in China (<https://weijing-rs.github.io/product.html>), the EBAS observation network in European countries (<https://ebas-data.nilu.no/Default.aspx>), and the IMPROVE observation network in the United States (<https://views.cira.colostate.edu/fed/Express/ImproveData.aspx>). Among them, a total of 20 points from EBAS and 142 points from IMPROVE were selected, along with 28 provincial capital cities in eastern China, for subsequent analysis. The following statistical analyses were conducted. As shown in Fig. S22, approximately three-quarters of the points were situated below the $y=x$ cutoff, indicating that the sulfate fractions of total $\text{PM}_{2.5}$ in the field observations or reanalysis datasets were higher than the simulations, suggesting that there was likely to be a missing sulfate formation pathway in the model simulation. The results by subregion in Fig. S23 also indicated that sulfate underestimation existed in China, European cities, and the United States, with the most significant underestimation observed in China. The aerosol aqueous-phase pathways that were focused on by this study could indeed contribute to sulfate formation, which might be responsible for the missing sulfate and be a potentiality for improving the performance of sulfate simulation.

For aerosol pH, we collected multi-year observation-constrained data and calibrated the simulation results, rather than limiting the data comparison to 2019. The spatial distribution represented the monthly average of aerosol pH, which was derived by the combination of the GEOS-Chem model and ISORROPIA II thermodynamic model (Fig. S24). The points in Fig. S24 denoted the pH reported in the published studies, which are collected and displayed on a public data platform Figshare (currently accessible private link: <https://figshare.com/s/f4d4c43c4a3c516d1b2c>, activated link after paper publication: <https://doi.org/10.6084/m9.figshare.24967032>). The pH values presented in this study demonstrated strong concordance with prior research, effectively reproducing both spatial differentials and seasonal fluctuations of aerosol pH. For regions such as Asia, Europe, and North America, where observations were abundant, the outcomes of comparative analyses were also favorable (Fig. S25), thus enhancing the reliability of our further calculations.

Aerosol liquid water content (AWC) was influenced by a combination of factors including aerosol concentration, relative humidity, and temperature. AWC could vary significantly in different regions, across different orders of magnitude. This variability would pose challenges for quantitative comparisons. Here, we aggregated a variety of studies relying on field observations and thermodynamic calculations for comparisons (currently accessible private link:

<https://figshare.com/s/f4d4c43c4a3c516d1b2c>, activated link after paper publication: <https://doi.org/10.6084/m9.figshare.24967032>), revealing that the reported extent of AWC was roughly consistent with our modeling results.

In summary, through comparative discussion, we believed that the PM_{2.5}, pH, and AWC in this study were all within reasonable limits. The potential importance of the sulfate formation in the aerosol aqueous phase was proved by comparing the percentages of sulfate, further illustrating the necessity of a global comprehensive simulation of assessing aerosol aqueous-pathway contributions.

Based on the valuable comments, the comparative analysis greatly improved the robustness of the work. We also added the comparative analysis to the revised manuscript and Supplementary information. Incorporating the sulfate formation pathways in AWC into GEOS-Chem could be the next work in future research to further refine the numerical simulation of sulfate. Thanks again for the valuable advice.

Page 13, Lines 314-326: Indeed, there could also be prospects that other aerosol-mediated reaction mechanisms reported recently could play key roles in sulfate formation, especially in polluted environments. Besides, uncertainties and deviations in the simulation of aerosol aqueous oxidations still exist due to the complexity of atmospheric multiphase chemistry. For instance, the important effects of aerosol ionic strength on certain reactions have been reported by experimental studies, like enhancing H₂O₂ oxidation rate while inhibiting TMI-catalyzed oxidation rate in deliquesced aerosol particles ¹, enhancing the multiphase oxidation rate of SO₂ by O₃ in aqueous acidified sea salt aerosols ², elevating the reaction rate constants for the oxidation of S(IV) by NO₂ in the aerosol water ³. Despite prior studies, more knowledge of the kinetics and thermodynamics in high ionic strength solutions is still needed for establishing solute strength-dependent kinetics, like tighter modeling and experimental constraints on kinetic parameters at different aerosol acidities and ionic strengths under conditions closer to the real atmosphere ^{4,5}.

Pages 16-17, Lines 426-439: For data calibration, the simulation results of PM_{2.5}, SO₄²⁻/PM_{2.5}, pH, AWC, and gaseous pollutants in this work are verified by comparing with the reanalysis datasets or observation-constrained datasets. The global reanalysis dataset “Satellite-derived PM_{2.5}” from the Atmospheric Composition Analysis Group at Washington University in St. Louis (<https://sites.wustl.edu/acag/datasets/surface-pm2-5/>, last access: 2024-3-31) is utilized (Fig. S20, S21). We also compare the concentration ratio of SO₄²⁻/PM_{2.5} between GEOS-Chem model simulations and field observations or reanalysis datasets (Fig. S22, S23), by gathering the publicly available data from the CHAP

reanalysis dataset for China (<https://weijing-rs.github.io/product.html>), the EBAS observation network in European countries (<https://ebas-data.nilu.no/Default.aspx>), and the IMPROVE observation network in the United States (<https://views.cira.colostate.edu/fed/Express/ImproveData.aspx>). Aerosol pH (Fig. S24, S25), AWC, and gaseous pollutants are verified by observation-constrained data from previous studies. The relevant datasets for reference and validation can be referred to <https://doi.org/10.6084/m9.figshare.24967032>.

Supplementary information Pages 28-31, Fig. S20-Fig. 25:

Fig. S20. Spatial distribution of $PM_{2.5}$ concentration from GEOS-Chem simulation and satellite-derived reanalysis dataset in January, April, July, and October 2019.

We utilized the reanalysis dataset “Satellite-derived $PM_{2.5}$ ” from the Atmospheric Composition Analysis Group at Washington University in St. Louis (<https://sites.wustl.edu/acag/datasets/surface-pm2-5/>, last access: 2024-3-31).

Fig. S21. Comparison of $\text{PM}_{2.5}$ concentration between GEOS-Chem simulation and satellite-derived reanalysis dataset in North Africa, Oceania, East Asia, Europe, and North America.

North Africa (5°N – 30°N , 10°W – 30°E), Oceania (10°S – 45°S , 110°E – 155°E), East Asia (5°N – 58°N , 100°E – 150°E), Europe (37°N – 70°N , 9°W – 34°E), and North America (25°N – 65°N , 60°W – 140°W) are focused for further comparison.

The relevant datasets for reference and validation can be referred to <https://doi.org/10.6084/m9.figshare.24967032> (currently accessible private link: <https://figshare.com/s/f4d4c43c4a3c516d1b2c>, activated link after paper publication: <https://doi.org/10.6084/m9.figshare.24967032>).

Fig. S22. $\text{SO}_4^{2-}/\text{PM}_{2.5}$ from GEOS-Chem simulation and observation in 2019.

We compared the concentration ratio of $\text{SO}_4^{2-}/\text{PM}_{2.5}$ between GEOS-Chem model simulations and field observations/reanalysis datasets. We gathered the publicly available data from the CHAP reanalysis dataset in China (<https://weijing-rs.github.io/product.html>), the EBAS observation network in European countries (<https://ebas-data.nilu.no/Default.aspx>), and the IMPROVE observation network in the United States (<https://views.cira.colostate.edu/fed/Express/ImproveData.aspx>). Among them, a total of 20 points from EBAS and 142 points from IMPROVE were selected, along with 28 provincial capital cities in China.

Fig. S23. Comparison of $\text{SO}_4^{2-}/\text{PM}_{2.5}$ between GEOS-Chem simulation and observation in East Asia, Europe, and North America.

The relevant datasets for reference and validation can be referred to <https://doi.org/10.6084/m9.figshare.24967032> (currently accessible private link: <https://figshare.com/s/f4d4c43c4a3c516d1b2c>, activated link after paper publication: <https://doi.org/10.6084/m9.figshare.24967032>).

Fig. S24. Aerosol pH from GEOS-Chem simulation and observation-based estimation in January, April, July, and October.

Aerosol pH from GEOS-Chem simulation in 2019 is represented as a spatial distribution map and observation-based estimation is represented as discrete points.

Fig. S25. Comparison of aerosol pH between GEOS-Chem simulation and observation-based estimation in East Asia, South Asia, Europe, and North America.

The relevant datasets for reference and validation can be referred to <https://doi.org/10.6084/m9.figshare.24967032> (currently accessible private link: <https://figshare.com/s/f4d4c43c4a3c516d1b2c>, activated link after paper publication: <https://doi.org/10.6084/m9.figshare.24967032>).

2) Authors mention in the abstract, and many other places, that rising H_2O_2 has prevented a steady decline in SO_4 . This may not be true from the observations. For example, in Beijing especially during heavy haze episode, large reduction in SO_2 emission has led to the change of dominated inorganic from sulphate to nitrate (Wang et al., 2020) and references therein. During these polluted episodes, although not dominate by sulphate, very high aerosol liquid water has been observed in Beijing up to $80 \mu\text{g}/\text{m}^3$. Not only Beijing, New Delhi also observed recorded high aerosol water up to $260 \mu\text{g}/\text{m}^3$ (Chen et al., 2022), which could contribute largely to aqueous phase SO_2 oxidation.

Response: Thanks for your valuable comment. We have made modifications to the relevant discussion. We carefully read and cited the literature (Wang et al., 2020, Chen et al., 2022), and described the effect of aerosol liquid water more clearly.

Aerosol liquid water is indeed an essential parameter facilitating haze development, and we also found the significant role of AWC (aerosol water content) in the exploration of the influencing factors of aerosol aqueous sulfate formation pathways in typical cities over the past two decades. We have described this part in the revised manuscript.

Besides, elevated H_2O_2 concentrations are probably inevitable on a global scale. Affected by the changes in tropospheric chemistry in recent years owing to the increase of CO, NO, NO_2 , VOCs, as well as the increasing UV-B radiation caused by the depletion of stratospheric ozone, the evidence from ice core

samples drilled at Summit Greenland showed a 60% increase in H₂O₂ concentrations during the last 150 years⁶; a prediction evaluated with a one-dimensional photochemical mode also suggested that H₂O₂ increased from 1980 to 2030 could be 100% or more in the urban boundary layer⁷. Sulfate aerosol concentration has decreased in recent years, and reduction might be further achieved by focusing on both the AWC and H₂O₂ concentrations.

Page 11, Lines 259-273: Sulfate concentrations in Chinese cities showed two-step declines from 2001-2009 and 2013-2019 (Fig. 3), which was closely associated with the fluctuation of H₂O₂ pathway contribution and its relevant factors. The previous step could be attributed to the decrease in aerosol water content (Fig. 4a). By providing more medium for multiphase reactions, increased AWC has been proven to accelerate secondary aerosol formation and trigger the positive feedback between water uptake and secondary formation in haze development, enhancing light scattering and reducing visibility, as well as suppressing the boundary layer height and worsening air pollutions^{8,9}. Compared with 2001, SO₂ emissions were enhanced in both 2005 and 2009, and the concentrations of the vital oxidant H₂O₂ were also increased (Fig. 4a). Nevertheless, SO₂ lacked an effective oxidation medium due to the reduced AWC^{10,11}. The positive benefits of the reduced AWC outweighed the negative impacts of SO₂ emissions and H₂O₂ concentrations, ultimately leading to lower sulfate concentrations during this period. The situation worsened in 2013 with conditions of high SO₂ emissions, high H₂O₂ concentrations, and high AWC, leading to a rebound in sulfate concentrations (Fig. 3).

3) Line 90-102. I feel it is more belong to the results. And here suddenly discuss about MBL sulphate is a bit distracting. I also do not understand that how could you infer recent 20 years conditions from simulation 2019?

Response: We appreciate your feedback. We have removed the distracting content and made changes for clarification. In this work, we explored the sulfate aqueous-phase formation pathways not only for 2019 but also for the last 20 years. We conducted the analysis over the last two decades based on the simulations from 2001, 2005, 2009, 2013, 2017, and 2019. Thanks for the helpful comment.

Page 4, Lines 89-95: In this work, we elucidate the significance of aqueous-phase sulfate formation pathways (i.e., H₂O₂, NO₂, O₃, TMI) in aerosol water based on a series of theoretical calculations utilizing fundamental data provided by the GEOS-Chem chemical transport model. We focus on the global spatiotemporal variabilities for January, April, July, and October 2019, both surface and vertical spatial scales. We also investigate the temporal trends of

aqueous sulfate formation pathways and the corresponding influencing factors in typical cities over 2001-2019 based on the simulations in 2001, 2005, 2009, 2013, 2017, and 2019.

4) Both NO₂ and O₃ show strong effect in high pH, and typically NO₂ and O₃ are highly correlated with each other. How do authors disentangle the effect of them? I suspect that if you turn off oxidation pathway of one of them, the contribution from the other one will increase, due to competition with the SO₂ oxidation. This might show some indirect contributions. However, this is just curious, H₂O₂ is the dominant factor anyhow.

Response: Thanks for your comment. We agree that NO₂ and O₃ are highly correlated with each other, and there may indeed be a competitive relationship between the two pathways theoretically, especially under alkaline conditions.

Here, in this work, we quantified the potential contribution of the aerosol aqueous pathways to global sulfate production based on component simulations from the GEOS-Chem benchmark model and subsequent theoretical calculations of formation rate. Methodologically, the calculation of NO₂ oxidation and the O₃ oxidation are independently conducted in this study.

To explore the sensitivity of these two pathways, here, we devised the following sensitivity experiment to investigate the potential relationship. We increased/decreased NO emissions by 20% and 50%, respectively, and mapped changes in the NO₂ pathway and O₃ pathway relative to the baseline scenario (Fig. R1, R2). On a global scale, the regions where increased and decreased in NO led to changes in the NO₂ pathway and the O₃ pathway were very limited. However, when examined at local and regional scales, the impact of changes in NO emissions was significant. Examples included regions such as the Sahara Desert region in North Africa, the North China Plain in East Asia, and the northern United States in North America. It is worth noting that due to the non-linear relationship between NO, NO₂, and O₃, there was not necessarily a positive correlation between changes in emissions and changes in pathway contributions.

Your advice is highly relevant, and the transformation of NO₂ and O₃ may play a crucial role, particularly for localized studies. Incorporating the four pathways into the GEOS-Chem model and conducting sensitivity tests on NO₂ and O₃ oxidation pathways can be the next endeavor in future studies, as per your recommendations.

Fig. R1. Impact of increased NO emissions on the NO₂ pathway and the O₃ pathway in January 2019. Changes in (a) NO₂ pathway and (b) O₃ pathway relative to the current baseline scenario for a 20% increase in NO emissions. Changes in (c) NO₂ pathway and (d) O₃ pathway relative to the current baseline scenario for a 50% increase in NO emissions.

Fig. R2. Impact of decreased NO emissions on the NO₂ pathway and the O₃ pathway in January 2019. Changes in (a) NO₂ pathway and (b) O₃ pathway relative to the current baseline scenario for a 20% decrease in NO emissions. Changes in (c) NO₂ pathway and (d) O₃ pathway relative to the current baseline scenario for a 50% decrease in NO emissions.

5) Line 168. Four modes? I do not understand it.

Response: Thanks for your comment. The term “four modes” refers to the four sulfate formation pathways under investigation, namely H₂O₂, NO₂, O₃, and TMI pathways. We have revised the phrasing, changing “modes” to “pathways”.

Page 7, Lines 169-172: Consistent with the surface results, the pathway distribution gave evidence that all four pathways contributed to sulfate formation considerably and were highly dependent on aerosol acidity, oxidants, and catalysts, at 950 mbar in January.

6) Line 240. Attributed to favorable meteorological conditions? Do authors expect that the meteorological conditions change largely between 2013-2019 and 2001-2009? If yes, please provide evidence.

Response: We appreciate you pointing it out. We meant to state that the previous step (2001-2009) of sulfate decline could be attributed to the decrease in aerosol liquid water content (AWC). It can be influenced not only by meteorological factors such as temperature and relative humidity but also by the hygroscopic properties of aerosols. We have revised the expression.

Page 11, Lines 261-262: The previous step could be attributed to the decrease in aerosol water content.

7) There are 7 megacities are chosen for more detailed analysis. Why these cities are chosen? I guess it is because of the availability of observational datasets from these cities and reasonable validation of the modelling outcomes. This links to my 1st point, more rigorous reference to previous observational studies are needed.

Response: We appreciate the helpful advice from you. We have added relevant descriptions of cities, as well as collected and summarized published data of relevant cities on a public data platform.

Considering the representativeness of different regions, we have chosen 7 urban areas mainly in East Asia, North America, Europe, and Oceania. These urban regions contain the capitals of 4 nations that are geographically distributed: Beijing (The People's Republic of China), Washington DC (The United States of America), London (The United Kingdom), and Canberra (The Commonwealth of Australia); as well as three prosperous megacities with high population density: Shanghai (China), Hong Kong (China), Los Angeles (USA). These regions have disparate meteorological conditions and atmospheric pollution levels, and the research in these areas focusing on the air quality and corresponding impacts on

human health has always received much scientific attention. We have added relevant descriptions.

Besides, there are indeed more available observational datasets from these megacities in focus, which can be used to compare and validate the reasonability of the modeling outcome. We have carefully reviewed previous observational studies and corresponding datasets and then compared our simulation results with the observation-constrained results, involving $\text{PM}_{2.5}$ concentration, ratio of $\text{SO}_4^{2-}/\text{PM}_{2.5}$, aerosol pH, aerosol water content, and gaseous pollutants concentration. In general, although there may be some uncertainties, the results of the modeling are basically reasonable. Detailed comparisons are referred to the response to the first comment. We have also added corresponding descriptions in the revised manuscript. Thank you for raising this point.

Page 9, Lines 226-229: Over the past two decades, the majority of the selected city areas (with disparate geographic locations, meteorological conditions, and atmospheric pollution levels) have exhibited a declining trend of SO_4^{2-} concentrations in January, with H_2O_2 oxidation emerging as the primary production pathway.

Page 15, Lines 391-395: To investigate temporal trends, we also conducted simulations in 2001, 2005, 2009, 2013, 2017, and 2019, and selected Beijing, Shanghai, Hong Kong, Washington DC, Los Angeles, London, and Canberra as representative cities for detailed discussion, considering the varied geographical locations, meteorological conditions, and air pollution levels.

Pages 16-17, Lines 426-439: For data calibration, the simulation results of $\text{PM}_{2.5}$, $\text{SO}_4^{2-}/\text{PM}_{2.5}$, pH, AWC, and gaseous pollutants in this work are verified by comparing with the reanalysis datasets or observation-constrained datasets. The global reanalysis dataset “Satellite-derived $\text{PM}_{2.5}$ ” from the Atmospheric Composition Analysis Group at Washington University in St. Louis (<https://sites.wustl.edu/acag/datasets/surface-pm2-5/>, last access: 2024-3-31) is utilized (Fig. S20, S21). We also compare the concentration ratio of $\text{SO}_4^{2-}/\text{PM}_{2.5}$ between GEOS-Chem model simulations and field observations or reanalysis datasets (Fig. S22, S23), by gathering the publicly available data from the CHAP reanalysis dataset for China (<https://weijing-rs.github.io/product.html>), the EBAS observation network in European countries (<https://ebas-data.nilu.no/Default.aspx>), and the IMPROVE observation network in the United States (<https://views.cira.colostate.edu/fed/Express/ImproveData.aspx>). Aerosol pH (Fig. S24, S25), AWC, and gaseous pollutants are verified by observation-constrained data from previous studies. The relevant datasets for reference and validation can be referred to <https://doi.org/10.6084/m9.figshare.24967032>.

8) line 289. “negative feedback of the increased H_2O_2 ”. And line 310, authors suggest for controlling of H_2O_2 . H_2O_2 is not primarily emitted, it comes from secondary formation which is highly non-linear. So, more detailed explanation of why do we have the negative feedback of the increased H_2O_2 , what are the major sources and pathways of H_2O_2 , which primary emission sources should be targeted in the control. The authors suggest VOCs are on an upward trend. This may not be true, at least not everywhere, e.g., in UK VOCs do keep decreasing every year (<https://www.gov.uk/government/statistics/emissions-of-air-pollutants/emissions-of-air-pollutants-in-the-uk-non-methane-volatile-organic-compounds-nmvocs>). Further discussion and clearer conclusive summary will improve the value of the paper and make more explicit and doable suggestions for the policymakers.

Response: Thanks for your valuable comments. We have added a conclusive summary and explicit suggestion in the revised manuscript.

As you mentioned, H_2O_2 is the vital photochemical secondary product closely related to atmospheric oxidizing capacity. H_2O_2 formation is a complex dynamic process that is intimately dependent on both the pollutant levels (NO , NO_2 , VOCs, CO , O_3 , and TMI) and meteorological parameters (light intensity, temperature, water vapor content). H_2O_2 generation is not only influenced by VOCs. It was indeed insufficient in the original manuscript to involve only elevated VOCs in the polluted atmosphere, thanks for pointing this out. To make it clearer, we explained the reason for the increased H_2O_2 concentration, reorganized the main sources of H_2O_2 , and discussed the possible H_2O_2 formation sources under various atmospheric situations. We also highlighted the necessity of controlling primary emissions including fossil fuel combustion, industrial processes, vehicle exhaust, mineral dust, and biomass burning. The details have been modified in the manuscript, and attached below.

Pages 13-15, Lines 334-377: Affected by the changes in tropospheric chemistry in recent years owing to the increase of carbon monoxide (CO), nitric oxide (NO), nitrogen dioxide (NO_2), volatile organic compounds (VOCs), as well as the increasing UV-B (ultraviolet radiation b) radiation caused by the depletion of stratospheric ozone, the evidence from ice core samples drilled at Summit Greenland showed a 60% increase in H_2O_2 concentrations during the last 150 years ⁶; a prediction evaluated with a one-dimensional photochemical mode also suggested that H_2O_2 increase from 1980 to 2030 could be 100% or more in the urban boundary layer ⁷.

As a vital photochemical secondary product, H_2O_2 generated from the binding of two HO_2 radicals can serve as an oxidant both in their own right and as a reservoir species for HO_x (OH and HO_2) radicals. OH can be produced via the photolysis of O_3 , nitrous acid (HNO_2), and aldehydes. HO_2 can be formatted

through the photo-oxidation of CO and VOCs by the OH radical, degradation of formaldehyde (HCHO) and other aldehydes by photolysis or by reaction with OH radical, the decomposition of peroxyacetyl nitrate (PAN), and the photodegradation of aromatic hydrocarbons¹². It's worth noting that the photochemical formation of H₂O₂ through HO₂ could be sensitive to ambient NO levels, due to the reaction of NO with HO₂ being faster than the bimolecular combination of HO₂, which could lead to substantial suppression of H₂O₂ formation via HO₂ if NO is abundant over a hundred ppt¹². This limitation effect has been reported in some areas, like Jungfrauoch Observatory in Switzerland¹³, Brittany in France¹⁴, and Hong Kong in China¹⁵. However, elevated H₂O₂ mixing ratios were also observed in some regions even during winter haze events with high NO and low HO₂ levels, like North China Plain¹⁶, implying other sources. Besides, observed H₂O₂ in the particle phase based on some urban sampling (Los Angeles, Beijing) was much greater than the concentration predicted by gas-particle partitioning from Henry's law, also indicating that the capability of generating H₂O₂ in aerosols, like redox chemistry of complexed transition metals and other redox-active species^{17,18}. For instance, field observations and laboratory experiments have proved that the photochemistry of H₂O₂ in-particle formation can be driven by transition metal ions (TMIs) and humic-like substances (HULIS) in deliquescent aerosols (RH>50%)¹⁶. Photochemical aging of atmospheric fine particles was also proved as a potential source for gas phase H₂O₂ under relatively dry conditions during daytime¹⁹. Another noteworthy source of gaseous H₂O₂ is the ozonolysis of alkene that is independent of radiation, which may be prominent in low photochemical conditions and may serve as the potential tropospheric source during autumn and winter at mid or high latitudes over the continents^{20,21} like Pabstthum in Germany (nighttime)²², Guangzhou in China (nighttime)²³, and Mt. Tai in China (nighttime)²⁴. Overall, H₂O₂ concentration in the atmosphere is dependent on the pollutant levels (CO, NO, NO₂, VOCs, O₃, TMI) and meteorological parameters (light intensity, temperature, water vapor content), intimately linking to O₃ and HO_x cycles, reflecting oxidation capacity of the troposphere. Further studies on the response of O₃ and H₂O₂ to NO_x (NO+NO₂) and VOCs as well as the interaction between secondary photochemical oxidants and aerosols are greatly needed, especially at various spatiotemporal scales. Attention to relevant anthropogenic emissions including fossil fuel combustion, industrial processes, vehicle exhaust, mineral dust, and biomass burning is imperative and control measures should be on the agenda, such as improving energy efficiency and supplying clean energy.

Method: mainly for clarification and improve the reproducibility.

1) accumulation mode of sea salt and dust are considered in the study, however, they are more abundant in the coarse model which is not considered. Please justify this.

Response: In this study, sea salt and dust were primarily utilized for calculating some species (Na^+ , Cl^- , K^+ , Ca^{2+} , Mg^{2+} , Mn^{2+} , and Fe^{3+}) in aerosol liquid water. On one hand, the aerosol itself was in the accumulation mode. On the other hand, the accumulation mode tended to absorb water, while the coarse mode contained almost no water. Therefore, we assumed that only the accumulation mode contributed to the ions in aerosols, while the coarse mode did not. In GEOS-Chem, the distinction between the accumulation mode (SALA) and coarse mode (SALC) of sea salt was made. In addition, we referred to Duncan Fairlie's suggestion that DST1 be regarded as the accumulation mode of dust, while DST2–DST4 were categorized as coarse-grained modes.

Dust reference:

https://wiki.seas.harvard.edu/geos-chem/index.php/Mineral_dust_aerosols

Sea salt reference:

https://wiki.seas.harvard.edu/geos-chem/index.php/Sea_salt_aerosols#Computing_PM2.5_concentrations_from_GEOS-Chem_output

2) Line 424-423. Please provide the references for the metal's speciation fractions.

Response: Thanks for the helpful comment. We have provided the references.

Page 16, Lines 402-405: The mass concentrations of some species are estimated based on the accumulation mode sea salt (SALA) and the accumulation mode dust (DST₁). $\text{Na}^+ = \text{SALA} \times 39.63\%$ ^{25,26}, $\text{Cl}^- = \text{SALA} \times 60.85\%$ ^{25,26}, $\text{K}^+ = \text{SALA} \times 1.1\%$ ^{25,26}, $\text{Ca}^{2+} = \text{DST}_1 \times 3\%$ ^{27,28}, $\text{Mg}^{2+} = \text{DST}_1 \times 0.36\%$ ²⁸, $\text{Mn} = \text{DST}_1 \times 0.063\%$ ^{29,30}, $\text{Fe} = \text{DST}_1 \times 2.425\%$ ^{29,30}.

3) Equation-4. Ionic strength. Where do you use it. I may miss that, but please explain after the eq.4, after you calculate the ionic strength, how do you use it for your analysis.

Response: We have described the related information in the revised manuscript. The previous study by Liu et al. suggested that the high solute strength of the aerosol particles significantly influenced the sulfate formation rate compared with the dilute solution ¹, for example, enhancing the H_2O_2 oxidation in the aerosol aqueous phase. Ionic strength calculated here is shown in Fig. S5, which displays the global spatial distribution of ionic strength as inferred from the

GEOS-Chem model and the ISORROPIA II model. Ionic strength may have an effect on sulfate formation through aqueous oxidation pathways, especially for the regions that had high ionic strength. Thank you for the advice.

Page 5, Lines 110-113: The importance of H₂O₂ pathway in aerosol multiphase chemistry is also supported by several recent studies⁵. An experimental research has demonstrated sulfate formation rate through H₂O₂ oxidation can be enhanced by the high ionic strength of aerosol particles (Fig. S5)¹.

Page 16, Line 416-420: The aerosol pH, aerosol water content (AWC), and ionic strength (I) play vital roles in sulfate aqueous formation, which are calculated by the ISORROPIA II model, a widely used thermodynamic equilibrium model with a high computational efficiency (<http://isorro피아.epfl.ch>)³¹. The model is run in the “forward mode” and “metastable state”. Aerosol pH (Fig. S6) and ionic strength (Fig. S5) are calculated.

4) Eq.6. Please provide the values and references for the Henry’s constants, and R value.

Response: Thanks for your comment. We provided the details in Supplementary Information Text S1 and Table S1. We also declared that in the manuscript.

Supplementary Information Page 3

Text S1. Calculations of aqueous-phase concentrations.

The dissolution of gas species (SO₂ and oxidants) in the aqueous phase follows Henry’s law.

$$[X(\text{aq})]=H_X \times P_{X(\text{g})} \quad (\text{S1})$$

where [X(aq)] is the aqueous-phase concentration of X species in equilibrium with P_{X(g)}, mol L⁻¹; H_X is Henry’s law coefficient, mol L⁻¹ atm⁻¹; and P_{X(g)} is the partial pressure of X in the gas phase, atm.

Henry’s constants and ionization constants are dependent on temperature. The temperature dependence of an equilibrium constant is given by the van’t Hoff equation³².

$$\frac{d\ln H_X}{dT} = \frac{\Delta H_X}{RT^2} \quad (\text{S2})$$

where ΔH_X is the enthalpy change at constant temperature and pressure; R is the molar gas constant, 8.314 J mol⁻¹ K⁻¹; T is the temperature, K. ΔH_X is a function of temperature, but it is approximately constant over small temperature ranges.

Thus,

$$H_T = H_{T_0} \exp\left[-\frac{\Delta H_{298K}}{R} \left(\frac{1}{T} - \frac{1}{T_0}\right)\right] \quad (\text{S3})$$

where T_0 can be 298K; H_{T_0} is the Henry's constant at 298K, mol L⁻¹ atm⁻¹. H_T is Henry's constant at a specific temperature. Table. S1 shows more details.

Table S1. Constants for calculating the aqueous-phase concentrations^{10,32}.

Gas species	Aqueous phase concentrations	H_{298K} (M atm ⁻¹) or K_{298K} (M)	$-\Delta H_{298K}/R$ (K)
	$[\text{SO}_2 \cdot \text{H}_2\text{O}(\text{aq})] = H_{\text{SO}_2} \times P_{\text{SO}_2}$	1.23	3145.3
SO ₂	$[\text{HSO}_3^-(\text{aq})] = K_{s1} \times [\text{SO}_2 \cdot \text{H}_2\text{O}(\text{aq})] / [\text{H}^+]$	1.3×10^{-2}	1960
	$[\text{SO}_3^{2-}(\text{aq})] = K_{s2} \times [\text{HSO}_3^-(\text{aq})] / [\text{H}^+]$	6.6×10^{-8}	1500
H ₂ O ₂	$[\text{H}_2\text{O}_2(\text{aq})] = H_{\text{H}_2\text{O}_2} \times P_{\text{H}_2\text{O}_2}$	1×10^5	7297.1
NO ₂	$[\text{NO}_2(\text{aq})] = H_{\text{NO}_2} \times P_{\text{NO}_2}$	1×10^{-2}	2516.2
O ₃	$[\text{O}_3(\text{aq})] = H_{\text{O}_3} \times P_{\text{O}_3}$	1.1×10^{-2}	2536.4

5) Eq.7-10. How do k1 ~ k8 been calculated? It seems not clear for me.

Response: Thanks for your comment. We provided the details in the Supplementary Information Text S2 and Table S2. We also declared that in the manuscript.

Supplementary Information Page 4

Text S2. The aqueous oxidation reaction of S(IV) into SO₄²⁻.

The aqueous-phase reaction equations of four oxidants are different.

For H₂O₂ oxidation pathway³²,

For NO₂ oxidation pathway³³,

For O₃ oxidation pathway³²,

For O_2 +TMI oxidation pathway^{32,34},

The corresponding rate expressions and rate constants for each pathway can be referred to Table S2.

Table S2. Rate expressions and rate constants of relevant aqueous reactions.

Oxidant	Reaction rate expressions ($R_{\text{oxidant}+\text{S(IV)}}$)	Rate constants (k)
H_2O_2	$k_1[\text{H}^+][\text{HSO}_3^-][\text{H}_2\text{O}_2(\text{aq})]/(1+K[\text{H}^+])$	$k_1 = 7.45 \times 10^7 \times e^{(-4430 \times (1/T - 1/298))} \text{M}^{-1} \text{s}^{-1}$ $K = 13 \text{M}^{-1}$
NO_2	$(k_2/[\text{H}^+] + K_{s2}k_3/[\text{H}^+]^2)K_{s1}[\text{SO}_2 \cdot \text{H}_2\text{O}(\text{aq})][\text{NO}_2(\text{aq})]$	$k_2 = 10^6 \text{M}^{-1} \text{s}^{-1}$ $k_3 = 10^{10} \text{M}^{-1} \text{s}^{-1}$
O_3	$k_4[\text{SO}_2 \cdot \text{H}_2\text{O}] + k_5[\text{HSO}_3^-] + k_6[\text{SO}_3^{2-}][\text{O}_3(\text{aq})]$	$k_4 = 2.4 \times 10^4 \text{M}^{-1} \text{s}^{-1}$ $k_5 = 3.7 \times 10^5 \times e^{(-5530 \times (1/T - 1/298))} \text{M}^{-1} \text{s}^{-1}$ $k_6 = 1.5 \times 10^9 \times e^{(-5280 \times (1/T - 1/298))} \text{M}^{-1} \text{s}^{-1}$
$\text{O}_2 + \text{TMI}$	$\begin{cases} k_7[\text{H}^+]^{-0.74}[\text{S(IV)}][\text{Mn(II)}][\text{Fe(III)}], \text{pH} \leq 4.2 \\ k_8[\text{H}^+]^{0.67}[\text{S(IV)}][\text{Mn(II)}][\text{Fe(III)}], \text{pH} > 4.2 \end{cases}$	$k_7 = 3.72 \times 10^7 \times e^{(-8431.6 \times (1/T - 1/297))} \text{M}^{-2} \text{s}^{-1}$ $k_8 = 2.51 \times 10^{13} \times e^{(-8431.6 \times (1/T - 1/297))} \text{M}^{-2} \text{s}^{-1}$

References:

Chen, Y., Wang, Y., Nenes, A., Wild, O., Song, S., Hu, D., Liu, D., He, J., Hildebrandt Ruiz, L., Apte, J. S., Gunthe, S. S., and Liu, P.: Ammonium Chloride Associated Aerosol Liquid Water Enhances Haze in Delhi, India, *Environmental Science & Technology*, 56, 7163-7173, 10.1021/acs.est.2c00650, 2022.

Wang, Y., Chen, Y., Wu, Z., Shang, D., Bian, Y., Du, Z., Schmitt, S. H., Su, R., Gkatzelis, G. I., Schlag, P., Hohaus, T., Voliotis, A., Lu, K., Zeng, L., Zhao, C., Alfarra, M. R., McFiggans, G., Wiedensohler, A., Kiendler-Scharr, A., Zhang, Y., and Hu, M.: Mutual promotion between aerosol particle liquid water and particulate nitrate enhancement leads to severe nitrate-dominated particulate matter pollution and low visibility, *Atmos. Chem. Phys.*, 20, 2161-2175, 10.5194/acp-20-2161-2020, 2020.

Reviewer #2

General Comments:

In this study, the authors show a quantification of the potential contribution of the aerosol aqueous pathways to global sulfate production using the GEOS-Chem model. Results display that sulfate in most of the continental surface areas (approximately 80%) were derived from H₂O₂ oxidation. It's very amazing. This work highlights the equal importance of managing both oxidants and precursors for effective sulfate control. I think this manuscript can be published after minor revision. There are several comments as following.

Response: Thanks sincerely for your insightful and supportive review of our manuscript. We greatly appreciate your time and thoughtful advice. We are delighted to hear that you found our study to be valuable, which emphasized the importance of managing both oxidants and precursors for effective sulfate control. We are committed to addressing your concerns and making necessary revisions to improve the clarity of our manuscript. We have carefully addressed these points in our revised manuscript and provided a point-by-point response below.

Main Comments:

1. The oxidation of SO₂ by OH is an important pathway for sulfate formation. Globally, this pathway contributes 20-27% to sulfate production (Alexander et al., 2009; Sofen et al., 2011). However, the OH pathway is not involved in GEOS-Chem model simulation. This pathway is expected to added to their simulation.

Response: We appreciate you pointing it out. We have supplemented OH oxidation in the revised manuscript and Supplementary Information.

As you mentioned, gas-phase oxidation through OH radical could also contribute to sulfate formation. We then added the simulation of the OH oxidation pathway that is included in the base model of GEOS-Chem, and the results are shown in Fig. S8. The formation rate of OH gas-phase oxidation was compared with that of the summed aerosol aqueous-phase oxidation, and the four sulfate formation pathways in aerosol water mentioned in this paper were derived from the combination of the GEOS-Chem simulations and subsequent theoretical calculations. The result indicated the OH pathway contributing to sulfate formation was comparable to the aerosol aqueous-phase pathway. The role of the OH pathway has been highlighted in numerous studies. What set this study apart was our emphasis on the importance of aerosol aqueous pathways on a global

scale.

Page 6, Lines 156-158: Besides, our calculations showed that the contribution of aerosol aqueous-phase oxidation was comparable to that of the OH gas-phase oxidation (Fig. S8).

Supplementary information Page 15, Fig. S8:

Fig. S8. Aerosol sulfate formation by OH gas-phase oxidation at the surface layer in 2019.

The spatial distribution of OH gas-phase oxidation rate was shown in (a) January, (b) April, (c) July, and (d) October. (e) Comparative statistics showed the comparison of sulfate formation rates for OH gas-phase oxidation and summed aerosol aqueous-phase oxidation.

The sulfate formation rates from OH gas-phase oxidation based on GEOS-Chem simulations were comparable with the sulfate formation rates of summed aerosol aqueous-phase oxidations based on the combination of the GEOS-Chem simulations and subsequent theoretical calculations.

2. In the introduction you mention the GEOS-Chem chemical transport model, but does that model address the above-mentioned areas of deficiency that you mention. This is puzzling. So the advantages of the model are not stated in the manuscript.

Response: Thanks for the insightful comment. We implemented the following improvements to the revised manuscript to clarify the novelty of our work and delineate the boundaries and framework of our study.

For sulfate concentration, the gap between traditional model simulations and observations still existed on a regional scale, especially during severe pollution in winter⁴, thus deriving scientific interest in other potential sulfate formation mechanisms other than cloud chemistry, like aerosol-mediated aqueous phase pathways. However, scientific knowledge on the relative importance of aerosol aqueous pathways at a global level is extremely limited⁵. Seeing that the temporal and spatial distributions of SO₂, oxidants, catalysts, and aerosol characters vary greatly worldwide, the expected dominant pathways would be variable in different regions and periods. Thus, the work conducted a global comprehensive simulation to assess aerosol aqueous-pathway contributions to offset the lack of comparability globally. It should be clarified at the outset that this paper utilized the GEOS-Chem benchmark model to provide the underlying data, on which theoretical calculations for sulfate formation in AWC were based. Incorporating the sulfate formation pathways in AWC into GEOS-Chem could be a next work in our future research to further refine the numerical simulation of sulfate.

The innovation of this study lies in the exploration of spatiotemporal variations of aerosol sulfate formation pathways (H₂O₂, NO₂, O₃, TMI) in the aqueous phase from a global GEOS-Chem simulation and provides recommendations for controlling sulfate from the perspective of aerosol aqueous-phase formation. The results indicated that the aqueous-phase pathways in aerosol water indeed made an undeniable contribution to sulfate formation globally, especially the H₂O₂ oxidation. The temporal trends of sulfate formation pathways and the corresponding influencing factors in typical cities in the recent 20 years also displayed that the increasing regional dominant oxidant H₂O₂ should be the key control. The lack of consideration of these aerosol aqueous-phase pathways in the traditional model might be a potential cause of model-observation gaps, and the involvement of these pathways might address the deficiency mentioned in the Introduction.

Page 2, Lines 24-26: Here, we quantify the potential contributions of the above-mentioned aerosol aqueous pathways to global sulfate production based on simulations from the GEOS-Chem model and subsequent theoretical

calculations.

Page 4, Lines 89-100: In this work, we elucidate the significance of aqueous-phase sulfate formation pathways (i.e., H₂O₂, NO₂, O₃, TMI) in aerosol water based on a series of theoretical calculations utilizing fundamental data provided by the GEOS-Chem chemical transport model. We focus on the global spatiotemporal variabilities for January, April, July, and October 2019, both surface and vertical spatial scales. We also investigate the temporal trends of aqueous sulfate formation pathways and the corresponding influencing factors in typical cities over 2001-2019 based on the simulations in 2001, 2005, 2009, 2013, 2017, and 2019. The aqueous-phase pathways in aerosol water could make undeniable contributions to sulfate formation globally. The abundance of the regional dominant oxidant, like H₂O₂, can be the key control. The understanding of sulfate formation regimes on global and regional scales can be useful for performing atmospheric oxidant control, implementing corresponding reduction policies, and further achieving sulfate mitigation.

3. In this study, their results are obtained only from the GEOS-Chem model simulation. The credibility of information in this study is a bit low due to the absence of observation data.

Response: We appreciate you for this thoughtful advice. We have added the model calibration and data comparison to address the robustness in the revised manuscript. The model results perform well compared with the observational datasets; and the aerosol aqueous-phase pathways can indeed contribute to sulfate formation, which could be a potentiality for improving the performance of sulfate simulation.

For PM_{2.5}, we conducted calibration by utilizing the reanalysis dataset “Satellite-derived PM_{2.5}” from the Atmospheric Composition Analysis Group at Washington University in St. Louis (<https://sites.wustl.edu/acag/datasets/surface-pm2-5/>, last access: 2024-3-31). This dataset estimated global PM_{2.5} concentrations using satellite, model, and ground-based monitoring data. Aerosol optical thickness from satellites such as MODIS, VIIRS, MISR, and SeaWiFS are combined with ground-based solar photometer (AERONET) observations to produce geophysical estimates. Statistical fusion methods further integrate additional PM_{2.5} measurements. The results are illustrated in Fig. S20. The spatial distribution of PM_{2.5} concentration in four months of 2019 obtained from GEOS-Chem simulations closely resembled that of the reanalysis dataset “Satellite-derived PM_{2.5}”. Moreover, we focused on North Africa (5°N–30°N, 10°W–30°E), Oceania (10°S–45°S, 110°E–155°E), East Asia (5°N–58°N, 100°E–150°E), Europe (37°N–70°N, 9°W–34°E), and North America

(25°N–65°N, 60°W–140°W) for further comparison (Fig. S21). Both the mean and median of GEOS-Chem simulations for these regions were acceptable, though slightly overestimated compared with the dataset.

For sulfate, the GEOS-Chem model utilized in this study accounted for sulfate formation just in the gas phase and cloud water phase. To validate the significance of sulfate formation in aerosol liquid water, we compared the concentration ratio of $\text{SO}_4^{2-}/\text{PM}_{2.5}$ between model simulations and field observations/reanalysis datasets. We gathered the publicly available data from the CHAP reanalysis dataset in China (<https://weijing-rs.github.io/product.html>), the EBAS observation network in European countries (<https://ebas-data.nilu.no/Default.aspx>), and the IMPROVE observation network in the United States (<https://views.cira.colostate.edu/fed/Express/ImproveData.aspx>). Among them, a total of 20 points from EBAS and 142 points from IMPROVE were selected, along with 28 provincial capital cities in eastern China, for subsequent analysis. The following statistical analyses were conducted. As shown in Fig. S22, approximately three-quarters of the points were situated below the $y=x$ cutoff, indicating that the sulfate fractions of total $\text{PM}_{2.5}$ in the field observations or reanalysis datasets were higher than the simulations, suggesting that there was likely to be a missing sulfate formation pathway in the model simulation. The results by subregion in Fig. S23 also indicated that sulfate underestimation existed in China, European cities, and the United States, with the most significant underestimation observed in China. The aerosol aqueous-phase pathways that were focused on by this study could indeed contribute to sulfate formation, which might be responsible for the missing sulfate and be a potentiality for improving the performance of sulfate simulation.

For aerosol pH, we collected multi-year observation-constrained data and calibrated the simulation results, rather than limiting the data comparison to 2019. The spatial distribution represented the monthly average of aerosol pH, which was derived by the combination of the GEOS-Chem model and ISORROPIA II thermodynamic model (Fig. S24). The points in Fig. S24 denoted the pH reported in the published studies, which are collected and displayed on a public data platform Figshare (currently accessible private link: <https://figshare.com/s/f4d4c43c4a3c516d1b2c>, activated link after paper publication: <https://doi.org/10.6084/m9.figshare.24967032>). The pH values presented in this study demonstrated strong concordance with prior research, effectively reproducing both spatial differentials and seasonal fluctuations of aerosol pH. For regions such as Asia, Europe, and North America, where observations were abundant, the outcomes of comparative analyses were also favorable (Fig. S25), thus enhancing the reliability of our further calculations.

Aerosol liquid water content (AWC) was influenced by a combination of factors including aerosol concentration, relative humidity, and temperature. AWC could vary significantly in different regions, across different orders of magnitude. This variability would pose challenges for quantitative comparisons. Here, we aggregated a variety of studies relying on field observations and thermodynamic calculations for comparisons (currently accessible private link: <https://figshare.com/s/f4d4c43c4a3c516d1b2c>, activated link after paper publication: <https://doi.org/10.6084/m9.figshare.24967032>), revealing that the reported extent of AWC was roughly consistent with our modeling results.

In summary, through comparative discussion, we believed that the PM_{2.5}, pH, and AWC in this study were all within reasonable limits. The potential importance of the sulfate formation in the aerosol aqueous phase was proved by comparing the percentages of sulfate, further illustrating the necessity of a global comprehensive simulation of assessing aerosol aqueous-pathway contributions.

Based on the valuable comments, the comparative analysis greatly improved the robustness of the work. We also added the comparative analysis to the revised manuscript and Supplementary information. Incorporating the sulfate formation pathways in AWC into GEOS-Chem could be the next work in future research to further refine the numerical simulation of sulfate. Thanks again for the valuable advice.

Pages 16-17, Lines 426-439: For data calibration, the simulation results of PM_{2.5}, SO₄²⁻/PM_{2.5}, pH, AWC, and gaseous pollutants in this work are verified by comparing with the reanalysis datasets or observation-constrained datasets. The global reanalysis dataset “Satellite-derived PM_{2.5}” from the Atmospheric Composition Analysis Group at Washington University in St. Louis (<https://sites.wustl.edu/acag/datasets/surface-pm2-5/>, last access: 2024-3-31) is utilized (Fig. S20, S21). We also compare the concentration ratio of SO₄²⁻/PM_{2.5} between GEOS-Chem model simulations and field observations or reanalysis datasets (Fig. S22, S23), by gathering the publicly available data from the CHAP reanalysis dataset for China (<https://weijing-rs.github.io/product.html>), the EBAS observation network in European countries (<https://ebas-data.nilu.no/Default.aspx>), and the IMPROVE observation network in the United States (<https://views.cira.colostate.edu/fed/Express/ImproveData.aspx>). Aerosol pH (Fig. S24, S25), AWC, and gaseous pollutants are verified by observation-constrained data from previous studies. The relevant datasets for reference and validation can be referred to <https://doi.org/10.6084/m9.figshare.24967032>.

Supplementary information Pages 28-31, Fig. S20-Fig. 25:

Fig. S20. Spatial distribution of $PM_{2.5}$ concentration from GEOS-Chem simulation and satellite-derived reanalysis dataset in January, April, July, and October 2019.

We utilized the reanalysis dataset “Satellite-derived $PM_{2.5}$ ” from the Atmospheric Composition Analysis Group at Washington University in St. Louis (<https://sites.wustl.edu/acag/datasets/surface-pm2-5/>, last access: 2024-3-31).

Fig. S21. Comparison of $PM_{2.5}$ concentration between GEOS-Chem simulation and satellite-derived reanalysis dataset in North Africa, Oceania, East Asia, Europe, and North America.

North Africa ($5^{\circ}N-30^{\circ}N$, $10^{\circ}W-30^{\circ}E$), Oceania ($10^{\circ}S-45^{\circ}S$, $110^{\circ}E-155^{\circ}E$), East Asia ($5^{\circ}N-58^{\circ}N$, $100^{\circ}E-150^{\circ}E$), Europe ($37^{\circ}N-70^{\circ}N$, $9^{\circ}W-34^{\circ}E$), and North America ($25^{\circ}N-65^{\circ}N$, $60^{\circ}W-140^{\circ}W$) are focused for further comparison. The relevant datasets for reference and validation can be referred to <https://doi.org/10.6084/m9.figshare.24967032> (currently accessible private link: <https://figshare.com/s/f4d4c43c4a3c516d1b2c>, activated link after paper publication: <https://doi.org/10.6084/m9.figshare.24967032>).

Fig. S22. $SO_4^{2-}/PM_{2.5}$ from GEOS-Chem simulation and observation in 2019. We compared the concentration ratio of $SO_4^{2-}/PM_{2.5}$ between GEOS-Chem model simulations and field observations/reanalysis datasets. We gathered the publicly available data from the CHAP reanalysis dataset in China (<https://weijing-rs.github.io/product.html>), the EBAS observation network in European countries (<https://ebas-data.nilu.no/Default.aspx>), and the IMPROVE observation network in the United States (<https://views.cira.colostate.edu/fed/Express/ImproveData.aspx>). Among them, a total of 20 points from EBAS and 142 points from IMPROVE were selected, along with 28 provincial capital cities in China.

Fig. S23. Comparison of $\text{SO}_4^{2-}/\text{PM}_{2.5}$ between GEOS-Chem simulation and observation in East Asia, Europe, and North America.

The relevant datasets for reference and validation can be referred to <https://doi.org/10.6084/m9.figshare.24967032> (currently accessible private link: <https://figshare.com/s/f4d4c43c4a3c516d1b2c>, activated link after paper publication: <https://doi.org/10.6084/m9.figshare.24967032>).

Fig. S24. Aerosol pH from GEOS-Chem simulation and observation-based estimation in January, April, July, and October. Aerosol pH from GEOS-Chem simulation in 2019 is represented as a spatial distribution map and observation-based estimation is represented as discrete points.

Fig. S25. Comparison of aerosol pH between GEOS-Chem simulation and observation-based estimation in East Asia, South Asia, Europe, and North America.

The relevant datasets for reference and validation can be referred to <https://doi.org/10.6084/m9.figshare.24967032> (currently accessible private link: <https://figshare.com/s/f4d4c43c4a3c516d1b2c>, activated link after paper publication: <https://doi.org/10.6084/m9.figshare.24967032>).

4. This study focused on sulfate formation pathways in continental regions. The transport of marine oxidants, precursors and aerosols to continents can have influences on the pathways of sulfate formation. Could the authors consider the transport impacts on sulfate formation?

Response: We appreciate your comment. In this work, we mainly focused on the aerosol aqueous sulfate formation pathways in continental regions. The impact of ocean transport on continents has been taken into account in the simulation results from the global three-dimensional GEOS-Chem chemical transport model. The transport influence has been involved in the calculations.

Besides, pollutant levels (CO, NO, NO₂, VOCs, O₃, TMI) in this century are intimately linked to the relevant anthropogenic emissions including fossil fuel combustion, industrial processes, vehicle exhaust, mineral dust, and biomass burning, despite transport of marine is likely to play roles in coastal areas. For instance, an observation study in Hong Kong (China) has shown higher H₂O₂ mixing ratios and pollution levels when large amounts of anthropogenic pollutants are transported from the Pearl River Delta economic zone (PRD)⁴⁶; in contrast, air masses came from the ocean with relatively low concentrations of primary pollutants, were associated with lower pollution levels and oxidant concentrations. Thus, in this work, we investigated aerosol aqueous sulfate formation pathways in continental regions that are more affected by anthropogenic emissions. The related aerosol aqueous pathways over the ocean and in the marine boundary layer is an interesting topic, which could be explored in future work.

5. Sulfate formation pathways in the haze events are quite distinct from the clean days. Such, the sulfate formation pathways in a city with serious air pollution are quite distinct from those in a city with low levels of aerosol pollution. However, results in this paper show that the H_2O_2 pathway is dominated the production of sulfate in Beijing, Shanghai, Hong Kong, Washington, and London, etc. The levels of aerosol pollution in these cities are quite different, but the sulfate formation pathways are quite similar. So, how do the authors explain this phenomenon?

Response: Thanks for your valuable comments. Among four aerosol aqueous formation pathways, H_2O_2 oxidation showed significant influence in worldwide sulfate formation according to the monthly average simulation in January, April, July, and October 2019.

For one thing, H_2O_2 concentration has been proven to increase worldwide in recent years by the evidence from ice core samples drilled at Summit Greenland that showed a 60% increase in H_2O_2 concentrations during the last 150 years⁶, affected by the changes in tropospheric chemistry owing to the increase of CO , NO , NO_2 , VOCs, as well as the increasing UV-B radiation caused by the depletion of stratospheric ozone. Thus, the elevated level of oxidant could provide a prerequisite for the H_2O_2 oxidation reaction, which can also be proved by Fig. 4.

For another thing, aerosols are generally acidic in these megacities ($\text{pH}<6$) that could provide similar reaction conditions: relatively higher acidity in North America and Europe, and lower acidity in Asia³², though the pollution levels vary due to regional differences in meteorological conditions and emissions. As sulfate formation is a pH-sensitive process, pathway contribution shows different features¹²: TMI+ O_2 pathway is more dominant in $\text{pH}<4$; the most obvious contribution from H_2O_2 oxidation is shown between pH 4 and 6; NO_2 oxidation is significant when $\text{pH}>5$; O_3 pathway plays a crucial role when $\text{pH}>7$. In our work, pH values are 5.06, 3.84, 2.49, 3.08, 2.38, 3.37, and 1.18 for Beijing, Shanghai, Hong Kong, Washington DC, Los Angeles, London, and Canberra, respectively, which are also almost consistent with the other observational research. Therefore, the acidic aerosols in these cities will provide appropriate conditions for the corresponding reactions, like H_2O_2 and TMI oxidations. While contributions from NO_2 and O_3 oxidation are limited since there are not many alkaline conditions in these megacities.

In brief, based on the increased H_2O_2 concentrations and appropriate acidic conditions, aerosol aqueous sulfate formation of these urban areas mentioned in this work tends to be mostly influenced by H_2O_2 oxidation. Besides, it's worth noting that in some local specific situations especially at intensive haze events,

sulfate formation in these highly polluted areas has also been proven to originate from other sources (in addition to aqueous-phase reaction), where it might go through some distinct processes compared with the cleaner regions.

6. Some areas should be explained more clearly. For example, why TMI has a stronger core effect in areas of higher aerosol acidity. Whereas most of the continental surfaces are about 80% hydrogen peroxide oxidized, the dominance of NO₂ oxidation is not obvious. The characteristics of the regions dominated by TMI, H₂O₂, and NO₂ could have been described (should have been a bit more clear like the O₃ description).

Response: Thanks for your valuable advice. We have incorporated additional details regarding the characteristics of regions dominated by O₃, TMI, NO₂, and H₂O₂ pathways.

O₃ oxidation is more predominant in desert areas with stronger alkalinity, due to O₃ can react most rapidly with SO₃²⁻ (alkaline promoted). Thus, the extent of areas dominated by the O₃ pathway largely aligns with the distribution of the central regions of the deserts.

TMI pathway plays a central role in sulfate formation over the regions with higher aerosol acidity and sufficient catalyst, due to the catalysts like Mn(II) and Fe(III) being more soluble in acidic conditions and the availability of catalysts would increase, suggesting more ability to oxidize S(IV). The regions where TMI pathways dominate tend to be in North Africa, West Asia, South Asia, and Oceania.

While NO₂ oxidation becomes more significant in weak acid conditions, which would play a pivotal role only when the aerosol pH was above 5. It was generally greater than the majority of worldwide pH values inferred from thermodynamic modeling, thus the prevalence of NO₂ oxidation is generally not widespread on a global scale, except for a small portion of North America and East Asia observed in January.

H₂O₂ is one of the most effective oxidants of S(IV), and the H₂O₂ oxidation reaction is very fast. The reaction process is near pH-independent, which is because the pH dependences of the S(IV) solubility and the reaction rate constant cancel each other, providing more chance to oxidize S(IV) over a wide range of aerosol pH. The remaining continental areas are predominantly dominated by the H₂O₂ pathway.

Overall, H₂O₂ oxidation was found to play the dominant role in aerosol aqueous sulfate formation over a wide range of pH, covering most continents. The NO₂

pathway was uncompetitive compared with other pathways. TMI and O₃ routes were of paramount importance in some localized areas (desert surroundings with high acidity and desert heartlands with high alkalinity, respectively). We have improved the corresponding descriptions in the manuscript.

Pages 4-5, Lines 105-110: The simulation indicated that most of the continental surface areas (approximately 80%) were dominated by H₂O₂ oxidation. H₂O₂ is very soluble in aqueous phase and is one of the most effective oxidants of S(IV). The near pH independence of the H₂O₂ oxidation reaction favors more chances for sulfate formation over a wide range of pH, which is caused by the fact that the pH dependences of the S(IV) solubility and the reaction rate constant cancel each other³².

Page 5, Lines 117-122: Aided by the increase in the solubility of transition metals under low aerosol pH conditions, the availability of catalysts in the aqueous phase would increase^{5,35}. O₂ oxidation catalyzed by soluble TMI, like Mn(II) and Fe(III), would play a central role in sulfate formation over the regions with higher aerosol acidity and sufficient catalysts, particularly in parts of North Africa, West Asia, South Asia, and Oceania, notably in April and July (Fig. S6).

Pages 5-6, Lines 131-138: In this simulation, the ascendancy of NO₂ oxidation was generally not evident on a global scale, except for a small part of North America and East Asia in January. Consistent with the past study¹⁰, NO₂ pathway would become pivotal only when the aerosol pH was above 5, which was generally greater than the majority of worldwide pH values inferred from thermodynamic modeling³⁶. Even under similar pH conditions, the H₂O₂ pathway also appears to have opportunities for oxidation, because of the pH-independence of reaction rate³². NO₂ pathway was less competitive compared with other pathways globally.

7. Giving a general conclusion based on the regional characteristics. The third point is that in my opinion some of the more important diagrams could be put into the body of the text, such as some of the important drivers (S14-S17), and given their corresponding explanations.

Response: We appreciate your valuable comment. We have refined sections S14-S17, integrated them into the revised manuscript Fig. 4, and summarized general conclusions. In these selected urban areas, precursors, oxidants, and aerosol water content are all important factors affecting aerosol aqueous sulfate formation rate changes, based on a more detailed analysis of influencing drivers.

Page 12, Lines 289-301: Alongside H₂O₂ oxidation, it's crucial to highlight the

significance of the TMI pathway, which played a pivotal role in some selected cities. Though TMI pathway in these years almost showed a downward trend compared with that in 2001, the attention should also be directed towards the elevated concentrations of Mn(II) and Fe(III), especially in Los Angeles, London, and Canberra (Fig. 4d). Moreover, despite the less importance of NO₂ and O₃ pathways, we also found some commonalities in pathway changes. The contributions from NO₂ and O₃ pathways generally declined in Washington DC, Los Angeles, London, and Canberra relative to 2001, and the change rates remained relatively stable. Conversely, there were significant interannual differences and apparent increases in Beijing, Shanghai, and Hong Kong, which is likely to be caused by the combined effects of change in the precursor, oxidants, pH, and AWC, wherein the increasing NO₂ levels and raising aerosol pH would take responsibility for the growth of NO₂ pathway; the raising aerosol pH would also be one of the non-negligible reasons for the increase in O₃ oxidation (Fig. 4b, 4c).

Fig. 4 | Temporal trends in oxidation rates and influencing factors for (a) H₂O₂, (b) NO₂, (c) O₃, and (d) TMI pathways in selected urban areas in 2005, 2009, 2013, 2017, and 2019, compared with that in 2001. The dot charts represent the change rate of oxidation rates (upper axis). The bar charts indicate the change of the influencing factors, shown by 100% stacked bar charts (lower axis).

8. In the introduction you mentioned that the main research focused on anthropogenic influences. However, there are relatively few analyses that mention anthropogenic activities in the discussion, so might you consider adding this part to the discussion. For example, involving anthropogenic activities in the future perspective at the end.

Response: Thanks for your helpful comment. Following your advice, we have added the relevant content to the discussion, including combing through the main sources of the important oxidant (hydrogen peroxide, H₂O₂) in detail and highlighting concerns about the associated anthropogenic emission sources.

Given that the continuous decline of SO₂ emissions is largely achieved on a global scale, attention to oxidants is highlighted for further sulfate reduction, like H₂O₂. Affected by the changes in tropospheric chemistry in recent years owing to the increase of CO, NO, NO₂, and VOCs, as well as the increasing UV-B radiation caused by the depletion of stratospheric ozone, the increasing H₂O₂ concentration has been proved by the evidence from ice core samples drilled at Summit Greenland showed a 60% increase in H₂O₂ concentrations during the last 150 years ⁶. To make it clearer, we explained the reason for the increased H₂O₂ concentration, reorganized the main sources of H₂O₂, and discussed the possible H₂O₂ formation sources under various atmospheric situations. We also highlighted the necessity of controlling primary emissions including fossil fuel combustion, industrial processes, vehicle exhaust, mineral dust, and biomass burning. Conclusive and explicit suggestions are summarized in the part of the discussion. The details have been modified in the manuscript, and attached below.

Pages 13-15, Lines 334-377: Affected by the changes in tropospheric chemistry in recent years owing to the increase of carbon monoxide (CO), nitric oxide (NO), nitrogen dioxide (NO₂), volatile organic compounds (VOCs), as well as the increasing UV-B (ultraviolet radiation b) radiation caused by the depletion of stratospheric ozone, the evidence from ice core samples drilled at Summit Greenland showed a 60% increase in H₂O₂ concentrations during the last 150 years ⁶; a prediction evaluated with a one-dimensional photochemical mode also suggested that H₂O₂ increase from 1980 to 2030 could be 100% or more in the urban boundary layer ⁷.

As a vital photochemical secondary product, H₂O₂ generated from the binding of two HO₂ radicals can serve as an oxidant both in their own right and as a reservoir species for HO_x (OH and HO₂) radicals. OH can be produced via the photolysis of O₃, nitrous acid (HNO₂), and aldehydes. HO₂ can be formatted through the photo-oxidation of CO and VOCs by the OH radical, degradation of formaldehyde (HCHO) and other aldehydes by photolysis or by reaction with OH radical, the decomposition of peroxyacetyl nitrate (PAN), and the

photodegradation of aromatic hydrocarbons¹². It's worth noting that the photochemical formation of H₂O₂ through HO₂ could be sensitive to ambient NO levels, due to the reaction of NO with HO₂ being faster than the bimolecular combination of HO₂, which could lead to substantial suppression of H₂O₂ formation via HO₂ if NO is abundant over a hundred ppt¹². This limitation effect has been reported in some areas, like Jungfraujoch Observatory in Switzerland¹³, Brittany in France¹⁴, and Hong Kong in China¹⁵. However, elevated H₂O₂ mixing ratios were also observed in some regions even during winter haze events with high NO and low HO₂ levels, like North China Plain¹⁶, implying other sources. Besides, observed H₂O₂ in the particle phase based on some urban sampling (Los Angeles, Beijing) was much greater than the concentration predicted by gas-particle partitioning from Henry's law, also indicating that the capability of generating H₂O₂ in aerosols, like redox chemistry of complexed transition metals and other redox-active species^{17,18}. For instance, field observations and laboratory experiments have proved that the photochemistry of H₂O₂ in-particle formation can be driven by transition metal ions (TMIs) and humic-like substances (HULIS) in deliquescent aerosols (RH>50%)¹⁶. Photochemical aging of atmospheric fine particles was also proved as a potential source for gas phase H₂O₂ under relatively dry conditions during daytime¹⁹. Another noteworthy source of gaseous H₂O₂ is the ozonolysis of alkene that is independent of radiation, which may be prominent in low photochemical conditions and may serve as the potential tropospheric source during autumn and winter at mid or high latitudes over the continents^{20,21} like Pabstthum in Germany (nighttime)²², Guangzhou in China (nighttime)²³, and Mt. Tai in China (nighttime)²⁴. Overall, H₂O₂ concentration in the atmosphere is dependent on the pollutant levels (CO, NO, NO₂, VOCs, O₃, TMI) and meteorological parameters (light intensity, temperature, water vapor content), intimately linking to O₃ and HO_x cycles, reflecting oxidation capacity of the troposphere. Further studies on the response of O₃ and H₂O₂ to NO_x (NO+NO₂) and VOCs as well as the interaction between secondary photochemical oxidants and aerosols are greatly needed, especially at various spatiotemporal scales. Attention to relevant anthropogenic emissions including fossil fuel combustion, industrial processes, vehicle exhaust, mineral dust, and biomass burning is imperative and control measures should be on the agenda, such as improving energy efficiency and supplying clean energy.

Reference

Alexander, B., Park, R. J., Jacob, D. J., Gong, S. 2009. Transition metal-catalyzed oxidation of atmospheric sulfur: Global implications for the sulfur budget. *Journal of Geophysical Research: Atmospheres*, 114(D2).

Sofen, E. D., Alexander, B., Kunasek, S. A. 2011. The impact of anthropogenic emissions on atmospheric sulfate production pathways, oxidants, and ice core $\Delta^{17}\text{O}$ (SO_4^{2-}). *Atmospheric Chemistry and Physics*, 11(7), 3565-3578.

References in the response

1. Liu, T., Clegg, S. L. & Abbatt, J. P. D. Fast oxidation of sulfur dioxide by hydrogen peroxide in deliquesced aerosol particles. *Proc. Natl. Acad. Sci. U.S.A.* **117**, 1354–1359 (2020).
2. Yu, C. *et al.* Ionic strength enhances the multiphase oxidation rate of sulfur dioxide by ozone in aqueous aerosols: Implications for sulfate production in the marine atmosphere. *Environ. Sci. Technol.* **57**, 6609–6615 (2023).
3. Chen, T. *et al.* Enhancement of aqueous sulfate formation by the coexistence of NO₂/NH₃ under high ionic strengths in aerosol water. *Environ. Pollut.* **252**, 236–244 (2019).
4. Su, H., Cheng, Y. & Pöschl, U. New multiphase chemical processes influencing atmospheric aerosols, air quality, and climate in the Anthropocene. *Acc. Chem. Res.* **53**, 2034–2043 (2020).
5. Liu, T., Chan, A. W. H. & Abbatt, J. P. D. Multiphase oxidation of sulfur dioxide in aerosol particles: Implications for sulfate formation in polluted environments. *Environ. Sci. Technol.* **55**, 4227–4242 (2021).
6. Anklin, M. & Bales, R. C. Recent increase in H₂O₂ concentration at Summit, Greenland. *J. Geophys. Res. Atmos.* **102**, 19099–19104 (1997).
7. Thompson, A. M., Owens, M. A. & Stewart, R. W. Sensitivity of tropospheric hydrogen peroxide to global chemical and climate change. *Geophys. Res. Lett.* **16**, 53–56 (1989).
8. Chen, Y. *et al.* Ammonium chloride associated aerosol liquid water enhances haze in Delhi, India. *Environ. Sci. Technol.* **56**, 7163–7173 (2022).
9. Wang, Y. *et al.* Mutual promotion between aerosol particle liquid water and particulate nitrate enhancement leads to severe nitrate-dominated particulate matter pollution and low visibility. *Atmos. Chem. Phys.* **20**, 2161–2175 (2020).
10. Cheng, Y. *et al.* Reactive nitrogen chemistry in aerosol water as a source of sulfate during haze events in China. *Sci. Adv.* **2**, e1601530 (2016).
11. Gao, J. *et al.* Targeting atmospheric oxidants can better reduce sulfate aerosol in China: H₂O₂ aqueous oxidation pathway dominates sulfate formation in haze. *Environ. Sci. Technol.* **56**, 10608–10618 (2022).
12. Lee, M., Heikes, B. G. & O’Sullivan, D. W. Hydrogen peroxide and organic hydroperoxide in the troposphere: a review. *Atmos. Environ.* **34**, 3475–3494 (2000).
13. Walker, S. J. *et al.* Processes controlling the concentration of hydroperoxides at Jungfraujoch Observatory, Switzerland. *Atmos. Chem. Phys.* **6**, 5525–5536 (2006).
14. Sauer, F., Limbach, S. & Moortgat, G. K. Measurements of hydrogen peroxide and individual organic peroxides in the marine troposphere. *Atmos. Environ.* **31**, 1173–1184 (1997).
15. Guo, J. *et al.* Atmospheric peroxides in a polluted subtropical environment:

- Seasonal variation, sources and sinks, and importance of heterogeneous processes. *Environ. Sci. Technol.* **48**, 1443–1450 (2014).
16. Ye, C. *et al.* Particle-phase photoreactions of HULIS and TMIs establish a strong source of H₂O₂ and particulate sulfate in the winter north China plain. *Environ. Sci. Technol.* **55**, 7818–7830 (2021).
 17. Xuan, X., Chen, Z., Gong, Y., Shen, H. & Chen, S. Partitioning of hydrogen peroxide in gas-liquid and gas-aerosol phases. *Atmos. Chem. Phys.* **20**, 5513–5526 (2020).
 18. Arellanes, C., Paulson, S. E., Fine, P. M. & Sioutas, C. Exceeding of Henry's law by hydrogen peroxide associated with urban aerosols. *Environ. Sci. Technol.* **40**, 4859–4866 (2006).
 19. Liu, P. *et al.* Photochemical aging of atmospheric fine particles as a potential source for gas-phase hydrogen peroxide. *Environ. Sci. Technol.* **55**, 15063–15071 (2021).
 20. Paulson, S. E. & Orlando, J. J. The reactions of ozone with alkenes: An important source of HO_x in the boundary layer. *Geophys. Res. Lett.* **23**, 3727–3730 (1996).
 21. von Salzen, K. *et al.* Sensitivity of sulphate aerosol size distributions and CCN concentrations over North America to SO_x emissions and H₂O₂ concentrations. *J. Geophys. Res. Atmos.* **105**, 9741–9765 (2000).
 22. Geyer, A. *et al.* Nighttime formation of peroxy and hydroxyl radicals during the BERLIOZ campaign: Observations and modeling studies. *J. Geophys. Res. Atmos.* **108**, (2003).
 23. Hua, W. *et al.* Atmospheric hydrogen peroxide and organic hydroperoxides during PRIDE-PRD'06, China: their concentration, formation mechanism and contribution to secondary aerosols. *Atmos. Chem. Phys.* **8**, 6755–6773 (2008).
 24. Ye, C. *et al.* Atmospheric hydrogen peroxide (H₂O₂) at the foot and summit of Mt. Tai: Variations, sources and sinks, and implications for ozone formation chemistry. *J. Geophys. Res. Atmos.* **126**, e2020JD033975 (2021).
 25. Sherwen, T. *et al.* Global impacts of tropospheric halogens (Cl, Br, I) on oxidants and composition in GEOS-Chem. *Atmos. Chem. Phys.* **16**, 12239–12271 (2016).
 26. Wang, X. *et al.* The role of chlorine in global tropospheric chemistry. *Atmos. Chem. Phys.* **19**, 3981–4003 (2019).
 27. Xuan, J. Emission inventory of eight elements, Fe, Al, K, Mg, Mn, Na, Ca and Ti, in dust source region of East Asia. *Atmos. Environ.* **39**, 813–821 (2005).
 28. Zhang, Y. *et al.* Modeling the global emission, transport and deposition of trace elements associated with mineral dust. *Biogeosciences* **12**, 5771–5792 (2015).
 29. Dong, X., Fu, J. S., Huang, K., Tong, D. & Zhuang, G. Model development of dust emission and heterogeneous chemistry within the Community Multiscale Air Quality modeling system and its application over East Asia. *Atmos. Chem. Phys.* **16**, 8157–8180 (2016).
 30. Tao, W. *et al.* Aerosol pH and chemical regimes of sulfate formation in aerosol water during winter haze in the North China Plain. *Atmos. Chem. Phys.* **20**,

- 11729–11746 (2020).
31. Fountoukis, C. & Nenes, A. ISORROPIA II: a computationally efficient thermodynamic equilibrium model for K^+ – Ca^{2+} – Mg^{2+} – NH_4^+ – Na^+ – SO_4^{2-} – NO_3^- – Cl^- – H_2O aerosols. *Atmos. Chem. Phys.* (2007).
 32. Seinfeld, J. H. & Pandis, S. N. Atmospheric Chemistry and Physics: From Air Pollution to Climate Change. (2016).
 33. Liu, T. & Abbatt, J. P. D. Oxidation of sulfur dioxide by nitrogen dioxide accelerated at the interface of deliquesced aerosol particles. *Nat. Chem.* **13**, 1173–1177 (2021).
 34. Ibusuki, T. & Takeuchi, K. Sulfur dioxide oxidation by oxygen catalyzed by mixtures of manganese(II) and iron(III) in aqueous solutions at environmental reaction conditions. *Atmos. Environ. (1967)* **21**, 1555–1560 (1987).
 35. Ye, C. *et al.* A critical review of sulfate aerosol formation mechanisms during winter polluted periods. *J. Environ. Sci.* **123**, 387–399 (2023).
 36. Pye, H. O. T. *et al.* The acidity of atmospheric particles and clouds. *Atmos. Chem. Phys.* **20**, 4809–4888 (2020).

Reviewer #1 (Remarks to the Author):

The authors have responded to all my comments and further improved the manuscript. Thanks for the efforts. I only have one further minor suggestion regarding the point-04 (the 'NO₂-O₃-H₂O₂' interlink one) in my previous comment. I think it is worthwhile to have a few sentences in the discussion or outlook to briefly talk about this question and suggestion for future study.

Reviewer #2 (Remarks to the Author):

The revised manuscript has addressed most of the comments raised in the first round. The revised manuscript can be accepted for publication.

A point-by-point response to the reviewer's comments is provided below. The original comments from the reviewer are in normal font, our replies are indented and in blue, and revised texts as they appear in the manuscript text are indented and in red.

REVIEWERS' COMMENTS

Reviewer #1 (Remarks to the Author):

The authors have responded to all my comments and further improved the manuscript. Thanks for the efforts. I only have one further minor suggestion regarding the point-04 (the 'NO₂-O₃-H₂O₂' interlink one) in my previous comment. I think it is worthwhile to have a few sentences in the discussion or outlook to briefly talk about this question and suggestion for future study.

Response: Thank the reviewer for taking the time to review our manuscript again. We appreciate the reviewer's positive comments and thoughtful advice. We have added the relevant content in the revised manuscript to highlight the interactive relationship between the precursors (NO₂-O₃-H₂O₂) and to point out the insight for future research (Page 15, Lines 365-378).

Page 15, Lines 365-378: Overall, H₂O₂ concentration in the atmosphere is dependent on the pollutant levels (CO, NO, NO₂, VOCs, O₃, TMI) and meteorological parameters (light intensity, temperature, water vapor content), intimately linking to O₃ and HO_x cycles, reflecting oxidation capacity of the troposphere. Given the close interlink between H₂O₂, NO₂, and O₃, the fluctuation of a certain oxidant is likely to affect other correlated oxidant levels, leading to indirect pathway contributions to sulfate production. Unravelling the interactive relationship between the precursors is also key to better understanding the sulfate formation mechanism. Further studies on the response of O₃ and H₂O₂ to NO_x (NO+NO₂) and VOCs as well as the interaction between secondary photochemical oxidants and aerosols are greatly needed, especially at various spatiotemporal scales. Attention to relevant anthropogenic emissions including fossil fuel combustion, industrial processes, vehicle exhaust, mineral dust, and biomass burning is imperative and control measures should be on the agenda, such as improving energy efficiency and supplying clean energy.

Reviewer #2 (Remarks to the Author):

The revised manuscript has addressed most of the comments raised in the first round. The revised manuscript can be accepted for publication.

Response: Thank the reviewer very much for the supportive review of our manuscript. These comments are valuable and helpful in revising and improving our paper. We greatly appreciate the reviewer's time and constructive advice.